Systematic shifts in Budyko relationships caused by groundwater storage changes
Laura E. Condon[1*] and Reed M. Maxwell[2]
[1]Department of Civil and Environmental Engineering, Syracuse University, Syracuse, NY
USA
[2]Integrated GroundWater Modeling Center, Department of Geology and Geological
Engineering, Colorado School of Mines, Golden CO USA
*Correspondence to: lecondon@syr.edu
**Abstract:**
Traditional Budyko analysis is predicated on the assumption that the watershed of
interest is in dynamic equilibrium over the period of study and thus surface water
partitioning will not be influenced by changes in storage. However, previous work has
demonstrated that groundwater surface water interactions will shift Buydko
relationships.  While Budyko approaches have been proposed to account for storage
changes, given the limited ability to quantify groundwater fluxes and quantity across
spatial scales, additional research is needed to understand the implications of these
approximations.  This study evaluates the impact of storage changes on Budyko
relationships given three common approaches to estimating evapotranspiration
fractions: (1) determining evapotranspiration from observations, (2) calculating
evapotranspiration from precipitation and surface water outflow, (3) adjusting
precipitation to account for storage changes.  We show conceptually that groundwater
storage changes will shift Budyko relationship differently depending on the way
evapotranspiration is estimated.  A one-year transient simulation is used to mimic all
three approaches within a numerical framework in which groundwater surface water
exchanges are prevalent and can be fully quantified.  The model domain spans the
majority of the Continental US and encompasses 25,000 nested watersheds ranging in
size from 100 $km^2$ to over 3,000,000 $km^2$.  Model results illustrate that storage changes
can generate different spatial patterns in Budyko relationships depending on the
approach used. This shows the potential for systematic bias when comparing studies
that use different approaches to estimate evapotranspiration. Comparisons between
watersheds are also relevant for studies that seek to characterize variability in the
Budyko space using other watershed characteristics. Our results demonstrate that
within large complex domains the correlation between storage changes and other
relevant watershed properties, such as aridity makes it difficult to easily isolate storage
changes as an independent predictor of behavior. However, we suggest that using the
conceptual models presented here comparative studies could still easily evaluate a
range of spatially heterogeneous storage changes by perturbing individual points to
better incorporate uncertainty in storage changes into analysis.
**1. Introduction:**
The Budyko hypothesis states that the fraction of precipitation (P) that leaves a
watershed through evapotranspiration (E), as opposed to runoff, can be predicted by
the aridity of the watershed [*Budyko*, 1958; 1974]. *Budyko* [1974] compared long-term
evapotranspiration fractions to aridity for 1,200 large watersheds around the globe and
showed that 90% of the variance in evapotranspiration ratio (E/P) could be described by
a single empirical curvilinear equation dependent only on aridity, often referred to as
the "Budyko Curve". Budyko noted that this consistent relationship is a reflection of the
dominance of macroclimate over large drainage areas and long time periods where it
can be assumed that a watershed is in steady state (i.e. when it can be assumed that
there are no storage changes over the period of analysis).

The simplicity of this relationship has since garnered much interest within the

hydrologic community for its potential to predict watershed behavior using only climate
variables, which are often easier to observe than many hydrologic variables, and
without relying on computationally expensive or heavily parameterized numerical
models.  In recent years, the Budyko hypothesis has also been put forward as a way of
predicting hydrologic sensitivity to climate change especially in ungauged basins [e.g.
*Donohue et al.*, 2011; *Jones et al.*, 2012; *Renner et al.*, 2014]. However, application of
this method has been partially limited by spatial variability between watersheds and the
required steady state assumption.

The original Budyko curve presented a universal relationship between

evapotranspiration and aridity [*Budyko*, 1974]. Subsequent work has shown that, while
the Budkyo curve is generally robust, climate alone is not sufficient to predict watershed
partitioning; the shape of the curve can vary between locations, especially for smaller
watersheds. Differences in behavior between river basins have been attributed to
seasonal lags in water and energy supply, vegetative and soil properties [*Donohue et al.*,
2007]. The original Budyko curve has been reformulated multiple times to incorporate
additional free parameters to reflect these differences [*Choudhury*, 1999; *Fu*, 1981;
*Milly*, 1994; *L. Zhang et al.*, 2001; *L. Zhang et al.*, 2004], and numerous studies have used
these modified formulations to relate curve parameters to physical basin characteristics
in many settings [e.g. *Li et al.*, 2013; *Shao et al.*, 2012; *Williams et al.*, 2012; *Xu et al.*,
2013; *Yang et al.*, 2009]. For example, *Li et al.* [2013] and *Yang et al.* [2009] evaluated
relationships between the shape of the Budyko curve and vegetation coverage.
Similarly, *Williams et al.* [2012] and *L. Zhang et al.* [2004] found distinct shape
parameters when comparing forested watersheds to grasslands, although it should be
noted that they reached the opposite conclusion about their relative magnitudes.
Others have focused on the role of soil moisture and noted differences in behavior
based on plant water availability and seasonal lags in supply and demand [e.g. *Milly*,
1994; *Yang et al.*, 2007; *Yokoo et al.*, 2008].

Many previous studies have demonstrated good predictive abilities using

modified Budyko formulations even when applied to smaller watersheds and shorter
time scales than those originally intended. However, poor performance in some
locations, especially over annual or seasonal time periods, has been attributed to the
influence of storage changes that violate the steady state assumption [*Milly and Dunne*,
2002; *Lu Zhang et al.*, 2008]. *Istanbulluoglu et al.* [2012] and *T Wang et al.* [2009]
showed interannual storage changes can produce a negative correlation between
evapotranspiration ratio and aridity that is counter to the Budyko curve for baseflow
dominated basins in the Nebraska Sand Hills.  *D Wang* [2012] evaluated inter-annual
storage changes for twelve watersheds in Illinois and showed that, on an annual
timescale, variability in runoff and storage is larger than evapotranspiration, and
accounting for storage can improve the performance of Budyko predictions. *Du et al.*
[2016] presented a method for explicitly accounting for storage changes within the
Budyko framework and demonstrated that this approach can greatly improve
performance in arid regions or over shorter time scales where the steady state
assumption is not valid.

These studies all indicate the potential importance of groundwater surface water

interactions within the Budyko framework and illustrate paths forward for incorporating
groundwater surface water interactions into Budyko analysis. However, the extensive
field work needed to fully quantify groundwater surface water exchanges is often not
possible and is counter to the simplicity and minimal data requirements of the Budyko
approach. Even in Budyko analysis focused on groundwater surface water interactions,
quantifying groundwater changes remains a limiting factor. For example, in some
studies, the impact of groundwater storage changes have been inferred from variability
around the Budyko relationship without directly measuring these changes [*Milly and*
*Dunne*, 2002; *Lu Zhang et al.*, 2008]. Others have addressed interactions more directly
using baseflow separation techniques that require only streamflow observations [*T*
*Wang et al.*, 2009] or lumped watershed models that parameterize baseflow and
recharge [*Du et al.*, 2016]. However, with both of these approaches the groundwater
system is still not directly simulated or observed.  *Istanbulluoglu et al.* [2012] and *D*
*Wang* [2012] did use observations of water table depth to directly quantify storage
changes and demonstrate the impact of this change within the Budyko framework; but
the study areas with this approach were relatively limited (four watersheds for
*Istanbulluoglu et al.* [2012], and twelve for *D Wang* [2012]).  Groundwater observations
sufficient to precisely characterize watershed storage changes are difficult to obtain and
are not widely available. Therefore, adding groundwater storage calculations into
Budyko analyses remains infeasible in many cases and more work is needed to
understand the sensitivity of Budyko relationships to changes in storage
There are three common approaches to estimate evapotranspiration (E) in
Budyko analysis (listed here in order of complexity): First, if E can't be measured
directly, it is often estimated as the difference between precipitation and river outflow
in a basin.  Second, E can be measured directly using a variety of field methods.  Third,
as is the case with the more recent studies that seek to account for storage changes,
observed E values can be augmented with measurements of groundwater surface water
exchanges to estimate the 'effective precipitation' that is available for surface processes
(i.e. outflow and E). Here we hypothesize that storage changes will bias Budyko results
in predictable ways, as has been indicated by previous studies, but that the direction of
the bias will vary based on the way that evapotranspiration is handled within a study.
We evaluate this hypothesis by comparing Budyko relationships generated following the
three different approaches using the outputs of a physically based hydrologic model
that directly simulates the integrated groundwater surface water system over a large
spatial domain at high resolution. The three primary goals of our comparative analysis
are as follows:

o  1. Evaluate the sensitivity of Budyko relationships to groundwater

storage changes

o  2. Characterize systematic differences in the impact of storage changes

on Budyko relationships

o  3. Illustrate variability between approaches across physical settings and

spatial scales

**2. Methods**
We use an integrated hydrologic model to simulate water and energy fluxes in both the
surface and the subsurface.  Here we apply a high resolution (1 km$^2$) simulation of the
majority of the continental U.S. which covers more than 6 M km$^2$ and simulates
hydrologic systems across a broad range of physical settings and storage change
magnitudes.  The model is driven using historical observed atmospheric forcings such as
precipitation and temperature and provides gridded outputs of all water and energy
fluxes throughout the system. We use simulated surface water flow, evapotranspiration
and groundwater surface water exchanges to calculate Budyko relationships using three
different approaches to estimate fluxes:
1. Calculating evapotranspiration from simulated runoff and precipitation
2. Using simulated evapotranspiration values directly
3. Using simulated evapotranspiration values directly and taking into account

storage changes.

Differences between the approaches are compared with storage changes in each basin
to evaluate the systematic impacts of these changes on Budyko relationships.

The numerical modeling approach used here provides several important

advantages for this type of analysis.  Within the model, groundwater surface water
exchanges for every watershed in the system are fully characterized. This guarantees
perfect closure of the water balance and means that we can mimic all three approaches
within a consistent numerical framework where storage changes are directly accounted
for. Furthermore, because the goal is to understand differences between approaches
and not to predict local Budyko parameters the key advantage here is the ability to
evaluate physically realistic behavior across a variety of physical settings and spatial
scales where groundwater can be fully accounted for. Within this context, it should also
be noted that the focus is on how groundwater storage changes perturb relationships.
Therefore, uncertainty in local model parameters is much less important than realistic
simulation of physical interactions for a range of storage changes and aridity values
within a controlled numerical framework.

Sections 2.1 and 2.2 detail the numerical modeling approach and the continental

scale simulation used for analysis.  An explanation of the source of each of the relevant
water balance terms generated from the model is provided in Section 2.3. Sections 2.4
and 2.5 explain the three different approaches for ET estimation and how they are
evaluated within the Budyko Framework.

**2.1 Hydrologic Modeling**
Previous work has evaluated the Budyko curve using hydrologic models of varying levels
of complexity. The abcd model employed by *Du et al.* [2016], among others, is a lumped
water balance model that includes baseflow and groundwater recharge using calibrated
parameters. *Yokoo et al.* [2008] used a different water balance model with a more
complex groundwater formulation that includes saturated and unsaturated zones, but
the authors noted limitations in simulating infiltration excess overland flow with this
approach. *Gentine et al.* [2012] applied a water balance model that includes a soil
bucket and can simulate infiltration excess overland flow; however it did not include
topography and was only applied at the plot scale. While these approaches do account
for storage in the subsurface and varying levels of complexity in groundwater surface
water exchanges, they all take a lumped approach and rely on calibrated parameters
that are not physically based.  The lumped parameter approach is illustrated in Fig. 1a.

Increasing in sophistication, *Troch et al.* [2013] used a semi-distributed model

that included shallow perched aquifers as well as root zone and soil moisture dynamics;
and *Koster and Suarez* [1999] evaluated a global circulation model that simulated land
surface and atmospheric processes using physically based equations. Incorporating
more sophisticated physical processes increases computational expense especially for
large high-resolution domains. To address this,  *Koster and Suarez* [1999] used a global
simulation but at low spatial resolution (4° by 5°), while *Troch et al.* [2013] limited their
analysis to the hillslope scale. Furthermore, both of these approaches are focused on
the land surface and shallow subsurface and neither included lateral groundwater flow
as shown in Fig. 1b.

To the authors' knowledge, no one has evaluated Budyko behavior over large

spatial scales using a hydrologic model that integrates lateral groundwater flow with
surface processes (Fig. 1c). So called integrated hydrologic models that incorporate
physically based lateral groundwater flow with overland flow and land surface processes
are a relatively new development in hydrologic modeling.  These tools are ideal for
capturing dynamic behavior and interactions throughout the terrestrial hydrologic cycle
and they have been increasingly applied over the last decade. To achieve this level of
complexity requires significant computational resources, and detailed model inputs.
These requirements have generally limited the application of integrated tools to
regional scale domains. Continental-scale high-resolution simulations have only recently
become technically feasible.
For this analysis, we use the first high-resolution integrated groundwater surface
water simulation of the majority of the continental US (CONUS) [*Maxwell and Condon*,
2016; *Maxwell et al.*, 2015]. The CONUS simulation was developed using the integrated
hydrologic model ParFlow-CLM [*Kollet and Maxwell*, 2006; 2008; *Maxwell and Miller*,
2005]. ParFlow simulates three-dimensional variably saturated groundwater flow using
Richards' equation:

$$S_s S(\psi_p) \frac{\delta \psi_p}{\delta t} + \phi \frac{\delta S(\psi_p)}{\delta t} = \nabla \cdot \left[ -K_s(x) k_r(\psi_p) \cdot \nabla(\psi_p - z) \right] + q_s \qquad (1)$$

where $S_s$ is the specific storage [$L^{-1}$], $S$ is the relative permeability [-], which varies with
pressure head $\psi_p$ [L] based on the *Van Genuchten* [1980] relationships, $t$ is time [T], $\phi$ is
the porosity of the subsurface [-], $K_s(x)$ is the saturated hydraulic conductivity tensor [LT$^-$
$^1$], $K_r$ is the relative permeability [-], which also varies with pressure head according to
the *Van Genuchten* [1980] relationships, $z$ is the depth below the surface [L] and $q_s$ is a
source/sink term [$T^{-1}$]. Note that units of $T^{-1}$ for the flux terms reflects the fact that they
are scaled by the cell thickness.
Overland flow is included in the groundwater flux term of Eq. (1) (i.e. in the first
term on the right hand side) using a free surface overland flow boundary condition that
applies continuity of pressure and flux across the boundary between the land surface
and the subsurface. Overland flow is solved using the kinematic wave approximation of
the momentum equation where the diffusion terms are neglected and it is assumed that
the bed slope, $S_o$ [-] is equivalent to the friction slope. Flow varies as a function of
ponded depth according to Manning's equation:

$$v = \frac{\sqrt{S_o}}{n} \, \psi_p{}^{2/3} \qquad\qquad (2)$$

where $n$ [TL$^{-1/3}$] is the Manning's roughness coefficient. Using this approach ParFlow is
able to solve variably saturated groundwater flow and overland flow simultaneously.
Practically this means that (1) the location of surface water bodies do not need to be
specified a priori and will develop wherever water ponds in the domain, and (2) two-
way groundwater surface water exchanges can evolve dynamically based on head
gradients and subsurface properties.

ParFlow is also coupled with a land surface model derived from the Common

Land Model (CLM) [*Dai et al.*, 2003]. In the combined ParFlow-CLM model [*Kollet and*
*Maxwell*, 2008], ParFlow solves the water balance in the subsurface and CLM solves the
combined water energy balance at the land surface.  At the land surface, the energy
balance ($R_{net}$) is comprised of sensible ($H$), latent ($LE$) and Ground ($G$) heat fluxes [Wm$^{-2}$]:

$$R_{net} = H + LE + G \qquad\qquad (3)$$


All of the energy fluxes listed in Eq. (3) vary with soil moisture. CLM uses

pressure head and saturation values for the upper subsurface layers (in this case the top
2m) simulated by ParFlow and passes infiltration fluxes back to ParFlow.  Land surface
processes are also driven by atmospheric forcing variables, which are provided as inputs
to the model. Forcing variables include short and longwave radiation, precipitation, air
temperature, atmospheric pressure, specific humidity and wind. Using these inputs,
CLM simulates multiple land surface processes including canopy interception,
evaporation from the canopy and the ground surface, plant transpiration, ground and
sensible heat fluxes as well as snow dynamics.

This study focuses on simulated evapotranspiration $E$ [LT$^{-1}$], which is the sum of

evaporation $Ev$, and plant transpiration $T$. CLM uses a mass transfer approach with
mean variables where evaporation is calculated using the gradient between the specific
humidity at the ground surface, $q_g$ [MM$^{-1}$], and the specific humidity at a reference
height, $q_a$ [MM$^{-1}$], scaled by a soil resistance factor $\beta$ [-], air density $\rho_a$ and the
atmospheric resistance, $r_d$ [-] as follows:

$$Ev = -\beta \rho_a \frac{q_g - q_a}{r_d}$$                (4)


The soil resistance factor is calculated based on the saturation relative to the residual
saturation and the saturation in the uppermost soil column (refer to *Jefferson and*
*Maxwell* [2015] for the complete formulation).

Similarly transpiration is calculated by scaling the potential evapotranspiration to

account for stomatal and aerodynamic resistance as follows

$$T = \left(R_{pp,dry} + L_w\right)L_{SAI}\left(\rho_a \frac{q_{sat} - q_a}{r_d}\right)$$                (5)


Here $R_{pp,dry}$ [-] is a scaling parameter, $L_w$ [-]is the fraction of the canopy that is covered in
water, $L_{SAI}$ is leaf and stem area index and $q_{sat}$ [-] is the saturated specific humidity [mm-
1]. $R_{pp,dry}$ is a function of light and moisture limitations. Parameters that are used to
determine leaf area index, reflectance and transmittance and root distributions vary by
land cover type and are provided as inputs to the model using the 18 land cover classes
defined by the International Geosphere Biosphere Program (IGBP).   For additional
details on the numerical approach and analysis on the sensitivity of evaporation and
transpiration within CLM the reader is referred to [*Ferguson et al.*, 2016; *Jefferson and*
*Maxwell*, 2015; *Kollet and Maxwell*, 2008; *Maxwell and Condon*, 2016].

**2.2 Model domain and simulations**
The analysis presented here is based on a previously developed transient ParFlow-CLM
simulation of the majority of the Continental US  (CONUS) documented in *Maxwell and*
*Condon* [2016]. The CONUS domain covers the majority of eight major river basins,
shown in Fig. 2 and spans roughly 6.3M square kilometers at 1km lateral resolution. The
integrated physically based approach employed for this simulation requires significant
computational resources. However, there are several key benefits that warrant this
costly approach; this simulation (1) provides high-resolution (1 km$^2$) gridded outputs
that fully define water and energy fluxes from the groundwater through the land surface
without calibration, (2) requires a minimal number of empirical parameters and (3)
directly simulates variably saturated lateral groundwater flow which has not been
incorporated in previous models used for Budyko analysis.

As detailed in in *Maxwell and Condon* [2016] and *Maxwell et al.* [2015], the

model extends 102 m below the subsurface with five vertical layers that contour to the
land surface using a terrain following grid formulation [*Maxwell*, 2013]. The vertical
resolution of the domain decreases with depth to better resolve the shallow subsurface.
Layer thicknesses are 0.1, 0.3, 0.6, 1 and 100m moving from the land surface down.
Spatially heterogeneous physical parameters for the subsurface include porosity,
saturated hydraulic conductivity and van Genuchten parameters. Subsurface spatial
units were determined using a national permeability map developed by *Gleeson et al.*
[2011] for the bottom 100 m of the domain and the soil survey geographic database
(SSURGO) for the top two meters. Maps of the subsurface units and their properties are
available in *Maxwell and Condon* [2016] and *Maxwell et al.* [2015].  The land surface was
derived from the Hydrologic data and maps based on the Shuttle Elevation Derivatives
at multiple Scales (HydroSHEDS) digital elevation model using a topographic processing
algorithm to ensure fully connected drainage network [*Barnes et al.*, 2016]. Vegetation
types were extracted from the USGS land cover dataset using the IGBP land cover
classifications.

The model was first initialized to a steady state groundwater configuration using

the ParFlow model without CLM starting from a completely dry domain and providing a
constant recharge forcing over the land surface to achieve a dynamic equilibrium.
Development of this steady state simulation and evaluation of the resulting
groundwater configuration are provided in [*Condon et al.*, 2015; *Maxwell et al.*, 2015].
Using the steady state groundwater configuration as a starting point, and following
some initialization period, the coupled ParFlow-CLM model was used to simulate the
fully transient system including land surface processes for water year 1985 (i.e. Oct. 1,
1984 through Sep., 30 1985), which was chosen as it is the most climatologically average
within the past 30 years. The transient simulation was driven by historical hourly
meteorological forcings for water year 1985 from the North American Land Data
Assimilation System Phase 2 (NLDAS 2) [*Cosgrove et al.*, 2003; *Mitchell et al.*, 2004].
Anthropogenic activities such as groundwater pumping and surface water storage are
not included in the transient simulation. Therefore the simulation represents natural
flows in a pre-development scenario, which is ideal for Budyko analysis. Complete
details of the development of the transient simulation are available in *Maxwell and*
*Condon* [2016].

The one-year simulation presented here intentionally violates the steady state

assumption. The purpose of our analysis is to evaluate the impact of net storage
changes on Budyko relationships, therefore a steady-state simulation is not the goal. It
can also be argued that storage changes will vary from year to year or depending on the
multi-year period analyzed. The 1985 simulation year is not presented as a prediction of
long-term storage variability, it is simply used to sample a range of groundwater surface
water exchange across variable climates and physical settings. We present a general
framework for understanding the impacts of storage changes in various Buydko
formulations using water year 1985 as a representative example.

Similarly, because we are focused on a comparative analysis within the Budyko

framework, the results are not dependent on local calibration between simulated
results and observations. The discrepancies between approaches stem from differences
in the variables used to create a water balance (refer to sections 2.3 and 2.4); these
findings are not sensitive to parameter uncertainty in the model. Still, the transient
simulation has been rigorously  validated against all publically available observations for
water year 1985. This includes transient observations at varying frequencies from 3,050
stream gauges, 29,385 groundwater wells and 378 snow stations for a total of roughly
1.2 million comparisons points. Flux tower observations were not available over this
period, but latent heat fluxes were also compared to the Modern Era Retrospective-
analysis for Research and Application (MERRA) dataset. Complete details of the model
validation are provided in the supplemental information of *Maxwell and Condon* [2016].

Although there are of course limitations to the model and significant uncertainties in

spatial model parameterization, especially for the subsurface, overall comparisons
between simulated and observed values demonstrate that the modeling approach is
robust. Stream-flow timing and magnitude are generally well matched in undeveloped
basins, snowpack timing and melt is accurate and spatial patterns in latent heat flux are
reasonable. Most importantly for this analysis, the model validation shows that ParFlow
is accurately capturing the relevant physical processes. Uncertainty in subsurface
parameterization, bias in atmospheric forcing data and lack of anthropogenic activities
were identified as key areas that could improve the local predictions of the model.
However, as discussed above, the purpose of this work is not to predict Budyko curve
parameters for water year 1985. The uncertainties listed here are therefore important
to note, but do not limit the utility of this tool as a test bed for evaluating interactions
across spatial scales and complex physical settings.

**2.3 Water Balance Components**
Outputs from the hydrologic simulation are used to quantify all of the relevant water
balance components for Budyko analysis.  Precipitation is an input to the ParFlow CLM
model. Within the model precipitation can infiltrate to the subsurface, contribute to
runoff or pond on the land surface. Evaporation occurs from ponded water, bare soil
and canopy interception. Additionally, roots pull water from the subsurface to support
transpiration for plants and lateral groundwater flow redistributes moisture within the
subsurface and can further support overland flow.  All of these processes occur within
every 1 km$^2$ grid cell in the domain. The focus of this work is on watershed function and
therefore the gridded results are aggregated to more hydrologically relevant units. The
domain is divided into 33,454 subbasins each containing a single stream. Subbasin
areas, outlined in Fig. 2, vary but are generally on the order of 100 km$^2$. The total
drainage area for every subbasin, henceforth referred to as the watershed, is defined by
tracing up the river network to encompass the entire upstream contributing area. This
results in 33,454 nested watersheds ranging in drainage area from about one hundred
square kilometers to over three million. For all of the following analysis we will focus on
the 24,235 watersheds that are contained within the highlighted regions of Fig. 2.
Similarly, while the simulation uses an hourly time step, here we evaluate annual values.

At the watershed scale, precipitation $P$ [L$^3$] is balanced by surface water

outflows, $Q_{out}$ [L$^3$], evapotranspiration, $E$ [L$^3$], and net groundwater surface water
exchanges, referred to as groundwater contributions, $G$ [L$^3$].

$$P = Q_{out} + E + G \qquad\qquad (6)$$

Equivalently this can be expressed in terms of ratios relative to incoming precipitation
where the sum of the outflow ratios sum to one:

$$1 = \frac{Q_{out}}{P} + \frac{E}{P} + \frac{G}{P} \qquad\qquad (7)$$

As noted above, every watershed fully encompasses its contributing area, and

therefore surface water inflows are zero. $P$ is the sum of the gridded annual
precipitation over the drainage area. Every watershed is defined to have a single outlet
point. $Q_{out}$ is the overland flow calculated hourly at the outlet using the ponded water
depth and Eq. (2) and summed over the simulation period. $E$ is the total evaporation and
transpiration simulated by ParFlow-CLM summed for every grid cell in the drainage area
over the year.

There are multiple ways to estimate groundwater contributions within the

model.  Using gridded model outputs, the exchanges across the boundaries of every
river cell can be summed to determine net contribution of groundwater to overland
flow. Similarly, we can aggregate hourly changes in groundwater storage for every sub
basin to determine total storage exchanges. Because we are interested in the net
contribution of groundwater to streamflow and evapotranspiration for this analysis, we
can take a simpler approach. Within our numeral framework we have guaranteed
closure of the water balance for every watershed and therefore the net change in
groundwater storage that contributes to the surface water budget is simply $P$ - $Q_{out}$ – $E$
based on Eq. (6). When calculated this way G encompasses the total groundwater
surface water exchanges (i.e. changes in storage) required to support the simulated
outflow and evapotranspiration. It should be noted that in this formulation G
encompasses both exchanges between groundwater and surface water, which can be
either positive fluxes from the surface to the subsurface or negative fluxes from
subsurface to the surface, as well as changes in surface water storage. The assumption is
that, over the annual simulation, changes in ponded water are small relative to
groundwater surface water exchanges and so we refer to G as simply groundwater
storage changes or groundwater contributions. We follow the convention that a positive
groundwater contribution denotes water that is infiltrating from the land surface to the
subsurface whereas a negative value indicates groundwater discharge which can either
occur from groundwater supported $E$ or baseflow contributions to streams.

This approach is focused solely on the net contribution of groundwater to the

surface water budget. Nested systems of local and regional lateral groundwater flow are
simulated within the model and previous work has evaluated spatial patterns and
physical drivers of lateral groundwater imports and exports across the domain [*Condon*
*et al.*, 2015; *Maxwell et al.*, 2015] as well as groundwater residence times [*Maxwell et*
*al.*, 2016]. Here we focus only on net exchanges with the surface that are relevant to the
Buyko formulation. We do not need to quantify lateral exchanges in the subsurface
directly for these purposes; however, it should be noted that the lateral redistribution of
groundwater that occurs within the model is still vital to generating realistic
groundwater configurations and supporting groundwater surface water exchanges.

In addition to the simulated evapotranspiration ($E$), potential evaporation $E_p$ is

calculated using Eq. (4), the hourly meteorological forcing data used to drive the
simulations (air temperature, atmospheric pressure, specific humidity, and wind speed),
and simulated ground temperatures in the uppermost layer of the model. To calculate
potential evaporation, as opposed to $E$, the $\beta$ parameter is set to one to eliminate soil
resistance and $q_g$ is the saturated specific humidity calculated based on the ground
temperature and atmospheric pressure.  As with $E$, hourly gridded $E_p$ values are
summed over the entire simulation period for every watershed drainage area. Using the
modeled simulated ground temperatures and model inputs to calculate $E_p$ ensures that
the $E_p$ values driven by the same water and energy inputs that control E in the
simulation.

Fig. 3 maps the aridity index ($E_p/P$) as well as each component of the water

balance from Eq. (7) expressed as ratios of precipitation. Subplots b and c show regional
trends in the relative importance of evapotranspiration as opposed to overland flow. In
the more arid western portions of the domain (shown in red on subplot a), $Q_{out}$ is small
compared to $E$ whereas in the more humid eastern portions of the domain (blue and
orange values in subplot a) the relative magnitude of $Q_{out}$ increases.

Within this annual simulation, subplot d shows that groundwater surface water

exchanges (G/P) can be a substantial portion of the water balance in much of the
domain. This  indicates that the system in not in steady state over the simulation period.
As discussed in Section 2.2 the one-year simulation time was intentionally selected for
this reason.  Here, we take advantage of the ability to directly calculate groundwater
surface water exchanges within a controlled numerical simulation where such
exchanges are prevalent in order to evaluate the impact of storage changes on Buydko
relationships across a range of spatial scales and climates.

The groundwater contribution ratio map also illustrates the importance of lateral

groundwater flow at multiple spatial scales within the system. Groundwater storage
gains (i.e. positive values of $G/P$) are prevalent in the western arid portion of the domain
and groundwater discharge to surface water is more common in the humid eastern
portion of the domain. Within large basins like the Missouri, positive groundwater
contributions occur in the headwater regions and transitions to negative values
downstream. This is an illustration of lateral groundwater convergence and regional
flow systems.  Note that results are mapped by subbasin, but all water balance
calculations are carried out for the complete watershed draining to a subbasin outlet.
Therefore, Fig. 3 should be viewed as a system of nested subbasins with values
representing progressively larger drainage areas as you move downstream. With this in
mind, it is also intuitive that some of the largest groundwater contribution ratios occur
in headwater basins while in downstream reaches on major rivers the values are smaller
indicating a regional balance between local groundwater surface water exchanges when
aggregating over larger drainage areas.

**2.4 Three approaches to evapotranspiration**

We have identified three common treatments of evapotranspiration within Budyko

analyses. As will be demonstrated later on, these three approaches are identical in
systems where the steady state assumption is valid and no storage changes are
occurring.  However, when this is not the case, we hypothesize that the different
formulations for evapotranspiration will yield systematically different results. Here we
summarize the three approaches to E and how each approach is mimicked within the
simulated results.

Precipitation and runoff are generally much easier to measure at the watershed

scale than evapotranspiration or groundwater storage changes.  As a result, in many
Budyko analyses evapotranspiration is not actually measured directly, but is calculated
as the difference between precipitation and surface outflow [e.g. *Greve et al.*, 2015;
*Jones et al.*, 2012; *Renner et al.*, 2014; *T Wang et al.*, 2009; *Xu et al.*, 2013; *Yang et al.*,
2009].  This approach relies on the assumption that changes in storage are negligible.
We refer to this as the *inferred evapotranspiration* approach and mimic it by
approximating the evapotranspiration ratio as simulated $(P- Q_{out})/P$. In other words, for
this approach, we disregard the simulated evapotranspiration values and generate a
new evapotranspiration estimate (i.e. the *inferred evapotranspiration*) indirectly from
the precipitation input to the model and the simulated overland flow. To be consistent
with other studies, we follow the standard assumption that storage changes are
negligible and do not include groundwater storage changes in this estimate. The
implications of this assumption are explored in the results section.

A more direct, if less common, approach is to quantify evapotranspiration from field

observations. This approach does not require a steady state assumption when
calculating evapotranspiration but it does require more rigorous field observations and
is therefore not feasible for Budyko analysis of data sparse areas. Within our simulation
results, however, 'data' is not a limitation. Our modeled outputs include gridded hourly
evapotranspiration for the entire domain. Simulated E values are aggregated by
watershed and used to represent the so called *direct evapotranspiration.* Note that in
this case we are still using simulated E not observations. The intention is to treat the
model as our synthetic truth and compare variations within this framework.

Finally, the most rigorous, and data intensive, approach is to quantify both

evapotranspiration and groundwater surface water exchanges directly. This approach
has been used in recent studies seeking to evaluate storage impacts on Budyko
relationships (e.g. *Istanbulluoglu et al.* [2012]; [*D Wang*, 2012]).  Changes in
groundwater storage are not used to adjust evapotranspiration values directly but they
can be applied to precipitation estimates to better reflect the quantity of water that is
available to partition into overland flow or evapotranspiration. This is defined as
effective precipitation and is calculated as precipitation minus groundwater contribution
(*P-G)*. The effective precipitation approach was used by *Du et al.* [2016] in their study of
Budyko relationships in arid basins. For this study we mimic the effective precipitation
approach by using the simulated (or *direct*) evapotranspiration and combining the
model input precipitation with the calculated groundwater contributions. The
adjustment for effective precipitation within the Budyko framework is covered in
Section 2.5.

It should be noted here that the first two approaches (i.e. inferred and direct

evapotranspiration) are commonly used in analyses that rely on the standard
equilibrium assumption while the final method is designed for situations where this is
not the case.  By comparing results between all three we consider the impact of nonzero
groundwater contributions both for approaches that assume it is negligible and that
account for it.

**2.5 Budyko analysis**

Budykyo's original formulation expressed evapotranspiration ratio (E/P) as a function of
aridity index ($E_p$/P) as follows [*Budyko*, 1974]:

$$\frac{E}{P} = \left\{ \frac{E_p}{P} \left[ 1 - \exp\left(-\frac{E_p}{P}\right) \tanh\left(\frac{P}{E_p}\right) \right] \right\}^{0.5} \tag{8}$$


Although the original analysis by Budyko did show some scatter around the curve, Eq.
(8) defines a universal relationship that does not include any free parameters to account
for spatial differences [*Budyko*, 1974]. Subsequent work has observed systematic
variability between watersheds that can be related to climate, land cover and soil
properties [e.g. *Donohue et al.*, 2007]. To reflect this, the original universal Budyko
formulation has been refined multiple times to include additional free parameters
[*Choudhury*, 1999; *Fu*, 1981; *Milly*, 1994; *L. Zhang et al.*, 2001; *L. Zhang et al.*, 2004]. For
a summary of these formulations refer to *Du et al.* [2016] and *L. Zhang et al.* [2004] .
Here we apply the commonly used Budyko formulation from *Fu* [1981] and *L.*
*Zhang et al.* [2004]:

$$\frac{E}{P} = 1 + \frac{E_p}{P} - \left(1 + \left(\frac{E_p}{P}\right)^\omega\right)^{1/\omega} \tag{9}$$


Eq. (9) includes one free parameter, $\omega$ which can range from one to infinity, henceforth
referred to as the shape parameter. $\omega$ is an empirical parameter that has not been
ascribed a specific physical meaning, but is generally conceptualized as an integrated
catchment property that reflects characteristics such as land cover, soil properties,
topography and seasonality [*L. Zhang et al.*, 2004].  If the evapotranspiration fraction
and the aridity index are both known, $\omega$ can be calculated for any point on a Budyko
plot using Eq. (9).
Fig. 4 plots Eq. (8) for a range of $\omega$ values. The bold line ($\omega$=2.6) is roughly
equivalent to the original Budyko equation (Eq. 8) [*L. Zhang et al.*, 2004]. Following the
original Budyko assumption of no change in storage, in humid locations where potential
evaporation is less than precipitation, the system is energy limited and the maximum
value of $E$ is $E_p$. Conversely, when the aridity index is greater than one the system is
water limited and the maximum $E/P$ value is one (indicating that all incoming
precipitation is evaporated).  As the shape parameter increases the curves moves
progressively closer to the water ($E/P$=1) and energy ($E/P=E_p/P$) limitations of the
system.
In the following sections, Budyko relationships are plotted and shape parameters
are evaluated for all three approaches using variations or Eq (9) as follows:
1.  Inferred  evapotranspiration: evapotranspiration is calculated from precipitation

and outflow so ($P$-$Q_{out}$)/$P$  is substituted for $E/P$ in Eq. (9).

2.  Direct evapotranspiration: Eq. (9) is applied as written.
3.  Effective precipitation: precipitation is replaced by effective precipitation (P-G)

which means  $E/(P\text{-}G)$ replaces E/P and $Ep/(P\text{-}G)$ replaces $Ep/P$ in Eq (9).


**3. Results and discussion**
Results and discussion are divided into two sections. In section 3.1 the three approaches
to evapotranspiration fractions are compared across the entire simulation domain.
Systematic differences are identified and evaluated as a function of groundwater
contributions. A conceptual framework is presented to explain the biases between
approaches. In Section 3.2 the potential implications of these differences are illustrated
by comparing spatial patterns between the three approaches as well as relationships
across spatial scales.

**3.1 The impact of storage changes on Budyko Relationships**
Fig. 5 plots every watershed in the domain shown in Fig. 2 using the three
approaches to estimate the evapotranspiration fraction. In all three figures the
watershed points follow the overlaid Budyko curves; 77% of the watersheds fall within
the 1.6 to 3.6 shape parameter lines for the inferred evapotranspiration approach, 51 %
for the direct approach and 72% for the effective precipitation approach. This
demonstrates that Budkyo relationships are recreated with the integrated hydrologic
model. However, there are some notable differences between methods. With the
inferred $E$ approach shown in subplot a, the points are focused near the water limit line
(i.e. $(P\text{-}Q_{out})/P=1$) for high aridity values. Conversely, with the direct approach (subplot
b), the evapotranspiration ratios are generally lower at high aridity values. Also, with the
direct approach, there are points with evapotranspiration ratios greater than one and
fall above the water limit. This would appear to violate the water balance and will be
discussed more later.

Systematic differences between the Budyko plots shown in Fig. 5  are explained

by the way groundwater contributions influence each approach. This is illustrated
conceptually in Fig. 6. in systems with groundwater surface water interactions, incoming
precipitation is equal to the sum of evapotranspiration, outflow and ground water
contributions (Eq. (6)). This means that the difference between precipitation and
outflow  will only equal evapotranspiration if there are no storage changes (i.e. G is
zero); if there are non-zero groundwater contributions then precipitation minus outflow
is actually a measure of  evapotranspiration plus groundwater contributions (and not
the intended evapotranspiration). In other words, instead of evaluating,

$$\frac{E}{P} = f\left(\frac{E_p}{P}\right) \tag{10}$$


as intended in the Budyko formulation, the inferred evapotranspiration approach shown
in Fig. 5a is actually plotting


$$\left(\frac{E}{P} + \frac{G}{P}\right) = f\left(\frac{E_p}{P}\right) \tag{11}$$


This is illustrated in Fig. 6a; the curve is now plotting the sum of the evapotranspiration
fraction and the groundwater contribution fraction, not the evapotranspiration fraction
for the original formulation shown in Fig. 4. The difference between the curve and the
limit lines in this case is still the outflow fraction though.

The direct evapotranspiration approach avoids the limitations of the inferred

approach by evaluating Budyo relationships as a function of the evapotranspiration
fraction as intended in Eq. (10). However, groundwater contributions will still bias the
results with this approach because the difference between precipitation and
evapotranspiration is outflow plus groundwater contribution (Eq. (6)). Thus, the curve in
Fig. 6b represents the evapotranspiration fraction (as with Fig. 4) but now the
partitioning is occurring between evaporation and runoff plus groundwater
contributions, not just runoff. This means that the maximum evapotranspiration fraction
(i.e. the upper water limit) is not one, but one minus the groundwater contribution
fraction.

This shift in the upper limits of water availability explains the values greater than

one in Fig. 5b; in these watersheds groundwater contributions are negative (i.e.
groundwater is supplying water to the land surface) and this allows for
evapotranspiration values that are greater than the incoming precipitation. Similar shifts
in the upper limits of the system for arid locations were found by *Potter and Zhang*
[2009] who noted that evapotranspiration was actually approaching a fixed portion of
potential evapotranspiration for high rainfall years in arid basins in Australia.

The effective precipitation approach is designed to maintain focus on

partitioning between evapotranspiration and overland flow by removing groundwater
contributions from the denominator of both ratios (i.e. adjusting both the x and y axes
in Fig. 6c) . This ensures that the modified outflow and evapotranspiration ratios will
sum to one even when groundwater surface water exchanges are occurring; to
accomplish this the modified ratios are expressed as a function of effective precipitation
not precipitation. It should also be noted from Fig. 6 that in the case where G is zero (i.e.
there are no storage changes), the three formulations are equivalent.
The systematic differences explained in Fig. 6 are evaluated by calculating the
shape parameter (Eq. (9)) for the curve corresponding to every watershed plotted in Fig.
5. Fig. 7 a-c plot the resulting shape parameters as a function of groundwater
contribution fraction colored by aridity  for each of the three approaches. Recall from
Fig. 4 that larger curve numbers fall closer to the upper limits on the Budyko plots and
positive groundwater contribution fractions occur when there is a net flux from the
surface water to the groundwater (i.e. net infiltration).  Positive G values are most
prevalent in the more arid western portions of the domain as is shown in Fig. 3d and
demonstrated by the shading in Fig. 7a-c where the most, red (arid) points occur further
to the right along the x axis. As would be expected from Fig. 6, Fig. 7. a-c illustrate
varying relationships between shape parameters and groundwater contributions for the
different approaches. Recall that all of the results are based on the same underlying
simulation so the differences in Fig. 7 result purely from accounting differences in how
the evapotranspiration fraction is calculated between approaches.
Both the inferred (6a) and direct approaches (6b) show clear, but contradictory,
relationships with groundwater surface water exchanges. There is a positive relationship
between the shape parameter and groundwater contribution fraction for the inferred
evapotranspiration approach at the lower limits of the system as delineated by the
dashed line in Fig. 7a. This indicates that in arid watershed watersheds, increased
groundwater contributions are correlated with larger evapotranspiration fraction (i.e.
with larger curve numbers). The behavior is consistent with Fig. 6a;  because the
groundwater contribution is included in the evapotranspiration fraction when
evapotranspiration is inferred from precipitation and outflow (i.e. $P\text{-}Q_{out} = E + G$),
nonzero groundwater contributions vertically shift points in the Buydko plot.
Taking this idea further, Fig. 7d shows that a constant positive groundwater
contribution applied across aridity values will vertically shift the Budyko curve relative to
a scenario with no storage changes if evapotranspiration is inferred. In the case of a
positive groundwater contribution, this vertical shift moves points closer to the water
and energy limits of the system and therefore increases their shape parameters.  Note
that in the Fu equation (Eq. (9)), Budyko curves with different shape parameter are not
parallel to one another and converge at low aridity values; therefore the same
groundwater contribution value changes the shape parameter differently depending on
the location within the Buyko plot.  The linear trend traced along the lower portion of
the scatter plot in Fig. 7a shows that for the lowest curve numbers, occurring in
watersheds with high aridity, there is a roughly linear relationship between
groundwater contribution and shape parameters. This approximate linearity occurs
because the Fu curves become almost parallel for high aridity values (see Fig. 4).  For
lower aridity values, this is not the case and the relationship between groundwater
contribution and shape parameter will be positive but nonlinear.

Fig. 7b plots groundwater contributions versus shape parameters similar to 6a

but for the direct evapotranspiration approach. Recall that with this approach the
groundwater contributions are now essentially lumped with the outflow fraction (as
opposed to the evapotranspiration fraction with the inferred approach, refer to Fig. 6a
and b).  This means that rather than shifting points vertically in the Budyko plot (i.e. Fig.
7d), positive groundwater contributions change the total water that is available for
evapotranspiration. This can be conceptualized as shifting the limits of how much total
water is available for evapotranspiration.

In this case a positive groundwater contribution (i.e. surface water infiltrating to

groundwater) is essentially a loss to the surface water system and decreases the upper
limit of water available to the system. Fig. 7e illustrates this point for a constant
groundwater contribution across the entire Buydko plot. When groundwater
contributions are present the upper water limitation on the system shifts from 1 to 1-
$G/P$ and the energy limitation shifts from $E_p/P$ to $E_p/P-G/P$.  However, if different
watersheds have varying levels of groundwater contribution this mean that each
watershed will now have a different upper limit; in other words, the evapotranspiration
fraction plus the outflow fraction is no longer always equal to one but rather one minus
the groundwater contribution fraction. This creates a nonlinear inverse relationship
between curve number and groundwater contributions. As the groundwater
contribution fraction increases, the decreasing upper bounds on evapotranspiration
fraction will bias the system towards lower curve numbers (refer to Fig. 4).  This is a
nonlinear relationship which can be shown by calculating the shape parameter as a
function of groundwater contribution fraction in Eq. (9) for the limiting case where there
is no outflow (i.e. $G/P=1-E/P$). The dashed line on Fig. 7b shows the resulting
relationship for a relatively high aridity value of 6.  The curve provides a good
approximation for the upper limit of Fig. 6b.

Finally, a scatter plot of shape parameters versus groundwater contribution

fraction for the effective precipitation case (Fig 6c) shows similar patterns with aridity
but no clear correlation between storage changes and shape parameters.  This is to be
expected because the effective precipitation approach  adjusts for groundwater
contributions in both the evapotranspiration ratios and the aridity index before plotting.
However, it should be noted that some dependence on groundwater contribution is still
to be expected the extent that groundwater surface water exchanges are also
correlated with other watershed properties. For example, groundwater contributions
levels can also be correlated with vegetation type, soil properties and other watershed
characteristics, which have been correlated to shape parameters in previous research
[e.g. *Li et al.*, 2013; *Shao et al.*, 2012; *Williams et al.*, 2012; *Xu et al.*, 2013; *Yang et al.*,
2009].

This is true for the other approaches too; while the effect of groundwater

contributions within each space can be precisely determined using Eq. (7) and Eq. (9), it
is important to note that the watersheds evaluated here are also heterogeneous in land
cover, topography and seasonality. Therefore, in the scatter plots shown in Fig. 7, the
relationships between shape parameters and groundwater contribution explained by
subplots d and e appear as limits rather than strong predictors.  This point is also made
by *Istanbulluoglu et al.* [2012] who evaluated the impact of groundwater storage
changes on Budkyo relationships using the inferred evapotranspiration approach and
adjusting for storage changes using estimates from groundwater observations. They
provide a similar conceptual model to Fig. 7d describing consistent shifts within the
Buydko space as a function of groundwater contribution. However, for the four basins in
Nebraska that they evaluated they found a negative relationship between inferred
evapotranspiration ratios and aridity. This was attributed to a strong negative
correlation between groundwater contribution fraction and aridity index. In other
words, for this subset of basins, they show that the resulting trend is controlled by the
dependence of groundwater contribution on other watershed characteristics.
Fig. 8 compares the shape parameters calculated with each approach to
illustrate the way that different assumptions can bias derived Budyko relationships. Fig.
8a shows the differences between the inferred and direct evapotranspiration
approaches, which are commonly used in studies that assume no change in storage.
Because groundwater contributions are incorporated into different components of the
water balance with these methods Fig. 8a shows that, for positive groundwater
contributions (green points), the inferred shape parameters are systematically higher
than the direct shape parameters, while the inverse is true when groundwater
contributions are negative (purple points). Furthermore, when groundwater
contributions are large (i.e. the dark green circles in subplot a), the direct method has
uniformly low shape parameters, but the inferred method still shows a range of shape
parameters. This is to be expected from the conceptual model of the direct
evapotranspiration approach (Fig. 7e) where we showed that high groundwater
contributions decrease the upper limit of the evapotranspiration ratio. This shift biases
the system towards uniformly low shape parameters that are less sensitive to other
watershed characteristics.
The direct and inferred evapotranspiration methods are also compared to the
effective precipitation approach, which does account for groundwater contributions
(Fig.7 b &c). As would be expected, the direct and inferred approaches have inverse
biases relative to the effective precipitation method; shape parameters are
systematically higher with the inferred approach relative effective precipitation and
lower for the direct approach. Here too the trends with groundwater contributions are
reversed with positive contributions creating a positive bias for the inferred case and a
negative bias for the effective precipitation case. This result is in keeping with the
conceptual model of groundwater contributions to each approach; with the inferred
evapotranspiration approach groundwater contributions are lumped with
evapotranspiration while in the direct approach they are lumped with outflows.

Also, there is a much stronger correlation between the inferred

evapotranspiration and effective precipitation approaches (Fig. 8b) than between direct
evapotranspiration and effective evapotranspiration approaches (Fig. 8c) ($r^2$ value of
0.96 comparing inferred vs. effective as opposed to 0.32 for inferred vs. direct).  This is
partially due to the lack of sensitivity of shape parameters in the direct approach when
groundwater contributions are large, as was previously noted and is also illustrated in
Fig. 8a. For all three cases, Fig. 8 demonstrates systematic variability in the shape
parameter even for relatively small groundwater contributions.  As with Fig. 7, Fig 8
there is still significant scatter in each of these comparisons. In this case the scatter is
caused by the fact that the shape parameter will be impacted (1) by how large the
groundwater contribution fraction is and (2) the aridity of the watershed. Groundwater
contributions shift points within the Budyko plot in a linear fashion (although the
direction varies according to the approach) but the resulting change in shape parameter
will have a nonlinear dependence on both aridity and evapotranspiration fraction.

**3.2 Spatial patterns and scaling**
Section 3.1 explored the relationship between groundwater storage and shape
parameters using the three different approaches to evapotranspiration fractions. Here,
we illustrate the impacts of these differences on spatial patterns in shape parameters
and scaling relationships. The intent is to provide a demonstration of how systematic
differences will propagate across spatial scales using the 1985 simulation as a test case.
Obviously local differences will vary depending on the time period used for analysis and
the associated levels of groundwater contribution.

Fig. 9 maps shape parameters for all of the roughly 33,000 nested watersheds in

the simulation domain calculated using the three different approaches to
evapotranspiration ratios. Even though the one-year transient simulation used for the
analysis presented does not meet the Budyko equilibrium criteria, Figs. 4c and 8c show
that realistic Budyko relationships are still found when groundwater contributions are
accounted for using the effective precipitation approach. *Xu et al.* [2013] built a neural
network model to predict shape parameters using long-term observations from 224
watersheds with drainage areas ranging from 100 to 10,000 $km^2$. They then predicted
shape parameters globally using a variety of catchment characteristics. Excluding the
small drainage areas with shape parameters greater than four, the spatial patterns
calculated here with the effective precipitation approach (i.e. the only approach that
corrects for groundwater contributions, Fig. 9c) match well with the global map
presented by *Xu et al.* [2013].

All three maps demonstrate local variability and regional trends in the shape

parameters. This spatial variability is partially caused by the spatial patterns in
groundwater contribution fraction shown in Fig. 3d; however, it is also a reflection of
variability in catchment characteristics such as vegetative properties, topography and
climate that have been correlated to Budyko relationships by previous studies[e.g. *Li et*
*al.*, 2013; *Milly*, 1994; *Shao et al.*, 2012; *Williams et al.*, 2012; *Xu et al.*, 2013; *Yang et al.*,
2009; *Yokoo et al.*, 2008]. The purpose here is not to isolate all of the sources of spatial
heterogeneity, rather to illustrate how spatial patterns change depending on the
treatment of storage.

Spatial patterns are consistent between the three approaches in the more humid

eastern portion of the domain, where groundwater contribution ratios are generally
smaller (Fig. 3d), but in the more arid western portion of the domain significant
differences are observed.  For both the inferred evapotranspiration and effective
precipitation approaches there are large red areas indicating shape parameters greater
than four where the evapotranspiration ratio is falling very close to the water limitation.
The areas with the highest shape parameters (i.e. greater than four) are generally
consistent between the inferred evapotranspiration and effective precipitation
approaches, but the inferred approach results in higher curve numbers throughout the
western portion of the domain than the effective precipitation approach. This is
consistent with Fig. 8b that showed strong correlations between the shape parameters
of these two approaches ($r^2$=0.96) but a slight positive bias with positive groundwater
contributions for the inferred evapotranspiration approach; 62% of watersheds overall
and 86% of watersheds with a positive groundwater contribution have a higher shape
parameter using the inferred evapotranspiration approach..

Conversely, with the direct evapotranspiration approach the western portion of

the domain has much lower shape parameters and less spatial variability. Again, this
finding is consistent with Fig. 7b and e, which show that when groundwater
contributions are high, the curve numbers are uniformly low because the flux from the
surface water system to the groundwater shifts the upper limit of the
evapotranspiration fraction down.  The systematic differences in Fig. 9, both with
respect to the shape parameter values and the spatial patterns in these parameters,
where groundwater surface water exchanges are occurring indicate the potential to
arrive at fundamentally different conclusions about spatial trends in shape parameters
depending on the approach used.

Next, we evaluate groundwater impacts a function of drainage area. Budyko

originally limited analysis to large basins (which he defined as drainage areas greater
than 10,000 km$^2$) where he argued that macroclimate can be expected to dominate
partitioning [*Budyko*, 1974]. Indeed subsequent work has shown that for smaller areas
vegetation dynamics become increasingly important [*Donohue et al.*, 2007].  Fig. 10
plots Budyko relationships for every watershed grouped by drainage area using the
effective precipitation formulation as an example.  In this figure, the drainage area is
increased from watersheds less than 1,000 km$^2$ (9a) to watersheds greater than 100,000
km$^2$ (9d).  This figure shows that the scatter decreases as drainage area increases and
the points converge around a single curve. This behavior illustrates increased
importance of local watershed characteristics for smaller drainage areas consistent with
previous studies [e.g. *Budyko*, 1974; *Donohue et al.*, 2007]. We do not show the other
two approaches for this example because similar convergence behavior with larger
drainage areas is found in all three cases.

The shape parameters estimated with the effective precipitation approach are

arguably the most comparable to other long-term studies that have assumed
equilibrium conditions (assuming that the watersheds they studied actually were in
equilibrium over the study period). The simulated median value found here is slightly
lower than the original Budyko value of 2.6 and the median value of 2.56 found by
[*Greve et al.*, 2015] using the 411 Model Parameter Estimation Experiment (MOPEX)
catchments in the US. However, it compares well with 1.8 median value for large
MOPEX basins in the US reported by *Xu et al.* [2013]; although, it should be noted that
*Xu et al.* [2013] report a higher 2.6 median value for small basins, and the median small
basin value reported found here is 2.0.  Part of this bias can likely be attributed to the
concentration of MOPEX basins in the eastern portion of the US where Fig. 9 shows that
shape parameters are generally higher. Overall, the consistency in spatial patterns and
convergence around the Budkyo curve for large drainage areas indicates that the
ParFlow-CLM model recreates Budyko relationships even over a relatively short annual
simulation period as long as groundwater contributions are adjusted for (i.e. using the
effective precipitation approach). However, for smaller watersheds variability in
catchment characteristics is still an important consideration.

While all three approaches have decreased variance with increased drainage

area, the median and variance are not necessarily consistent between methods. Fig. 11
shows the interquartile range of shape parameters for each approach with increasing
drainage area.  In all three cases, the 75$^{th}$ percentile shape parameters decrease and the
25$^{th}$ percentile shape parameter increases with increasing area. Again this indicates
increased importance of watershed characteristics at smaller scales; local variability is
muted and the probability of observing very high or very low shape parameters
decreases as the scale increases from smaller to larger watersheds.  In the case of the
inferred and direct evapotranspiration approaches, because groundwater contributions
are not accounted for in the calculations, some of this variability can also be attributed
to spatial patterns in groundwater surface water exchanges and lateral groundwater
flow. As previously noted, the groundwater contribution map (Fig. 3d) shows that the
largest, positive or negative, groundwater contribution fractions generally occur in small
headwater basins. Across larger areas, local groundwater surface water exchanges
balance out and the overall groundwater contribution fractions for large watersheds
tend to be smaller.

Consistent with Figs. 7 and 8, the inferred evapotranspiration and effective

precipitation approaches are the most similar. For the largest drainage areas, the
median shape parameter is 1.8 for the inferred evapotranspiration approach, 1.5 for the
direct evapotranspiration approach and 1.7 using effective precipitation. The direct
evapotranspiration formulation has systematically lower shape parameters than the
other two approaches; the median value for this method is consistently below the other
two. Again this agrees with section 3.1 where we demonstrated an inverse relationship
between shape parameters and groundwater contributions. The direct
evapotranspiration approach also has a consistently smaller interquartile range than the
other two methods. This results from the negative correlation with groundwater
contribution and the decreased sensitivity that was shown for small shape parameters
in arid locations. Fig. 11 shows that all three approaches will yield qualitatively similar
scaling relationships and convergence for large basins; however, the shape parameter
values will vary.

**4. Conclusions**

One of the primary assumptions of the Buydko hypothesis is that watersheds are

in equilibrium and there are no changes in storage. This means that all incoming
precipitation will either leave the watershed as evapotranspiration or overland flow.
While the original Budyko curve has been well verified with observations from around
the globe, it is also now widely accepted that the relationship between
evapotranspiration ratios and aridity indices is not universal and some additional curve
parameters are needed to account for spatial variability between watersheds. Many
subsequent studies have related curve parameters to catchment properties such as
vegetation, topography and seasonality [e.g. *Li et al.*, 2013; *Shao et al.*, 2012; *Williams et*
*al.*, 2012; *Xu et al.*, 2013; *Yang et al.*, 2009]. More recently, additional studies have
shown that if groundwater surface water exchanges are present this can also influence
the shape of the curve and account for additional variability between watersheds [*Milly*
*and Dunne*, 2002; *Lu Zhang et al.*, 2008].
While methods have been developed to account for storage changes within the
Budyko framework [e.g. *Du et al.*, 2016], very few studies have sufficient data on
groundwater surface water interactions to evaluate the validity of the equilibrium
assumption, much less to precisely quantify storage changes in their analysis. One of the
key advantages of the Budyko approach is its ability to predict behavior based on a small
number of relatively easy to obtain observations.  Given its common application to data
sparse watersheds, where even evapotranspiration measurements are often not
available, directly quantifying groundwater surface water exchanges in these locations
seems unlikely.  Therefore,  it is important to understand the sensitivity of Budyko
relationships to uncertainty in storage changes in a general context that can be used to
interpret results were precise measurements are not available.
Previous work has demonstrated systematic shifts in Budyko plots caused by
groundwater surface water interactions [*Du et al.*, 2016; *Istanbulluoglu et al.*, 2012;
*Milly and Dunne*, 2002; *D Wang*, 2012; *L. Zhang et al.*, 2004]. Here we demonstrate that
the influence of groundwater storage changes on Budyko results will vary depending on
how evapotranspiration is handled in the study. If evapotranspiration is measured
directly, positive groundwater contributions (i.e. net infiltration from the surface to the
subsurface) shift shape parameters down; conversely, if evapotranspiration is estimated
using precipitation and runoff positive groundwater contributions will increase shape
parameters. In both cases the sensitivity of the shape parameter to storage changes
varies non-linearly with both the aridity of the watershed and the evapotranspiration
fraction.
Using a one-year simulation with an integrated hydrologic model we
demonstrate these differences can result in different conclusions about spatial patterns
in Budyko relationships and the median shape parameter across spatial scales. This
indicates that it is important to consider the approach used for estimating
evapotranspiration fractions when comparing results between studies, and provides a
demonstration of the types of bias that would be expected if different methods are
used.
These results also have implications for the myriad of studies that seek to relate
shape parameters for Buydko curves to other watershed characteristics. The conceptual
models shown here illustrate that groundwater contributions will shift points in
consistent and predictable ways when other variables are held constant (i.e. if you apply
a consistent groundwater contribution across the entire range of aridity values or
consider the shift of a single point with a given aridity value).  However, we use the
results from our integrated hydrologic model to demonstrate that that within complex
heterogeneous domains groundwater surface water exchanges are spatially
heterogeneous and depend on  watershed characteristics such as aridity values,  which
can also influence Budyko relationships. The scatter in Figs. 6 and 7 demonstrate that
groundwater contributions cannot easily serve as an independent predictor of the shape
of Budyko relationships.  This also shows that in large comparative studies, the bias
caused by groundwater surface water interactions may not be readily apparent because
it will vary from watershed to watershed.
The intention of these comparisons is not to discredit previous approaches,
rather to illustrate the potential impacts of assuming equilibrium conditions across a
broad range of physiographic settings and spatial scales without the ability to verify this
assumption. Our results show that even when changes in storage are occurring, large
watersheds still roughly follow Budyko curve; however the shape parameter and scatter
will vary with groundwater contribution and depending on how evapotranspiration is
quantified.   We suggest that studies that cannot verify the equilibrium assumption
using groundwater observations include additional analysis to evaluate the sensitivity of
their findings to uncertainty in storage changes by perturbing points using the
conceptual models presented here.  Even if groundwater contributions cannot be
directly incorporated into analyses, this can help determine whether differences in
shape parameters are actually resulting from unique basin characteristics or uncertainty
in storage.

**Data Availability:**
All data from this analysis are available upon request. Instructions for accessing the
ParFlow simulations used here are provided in [*Maxwell et al.*, 2016].

**Acknowledgements:**
Funding for this work was provided by the US Department of Energy Office of Science,
Offices of Advanced Scientific Computing Research and Biological and Environmental
Sciences IDEAS project. The ParFlow simulations were also made possible through high-
performance computing support from Yellowstone (ark:/85065/d7wd3xhc) provided by
National Center for Atmospheric Research's Computational and Information Systems
Laboratory, sponsored by the National Science Foundation.

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

**Figures:**

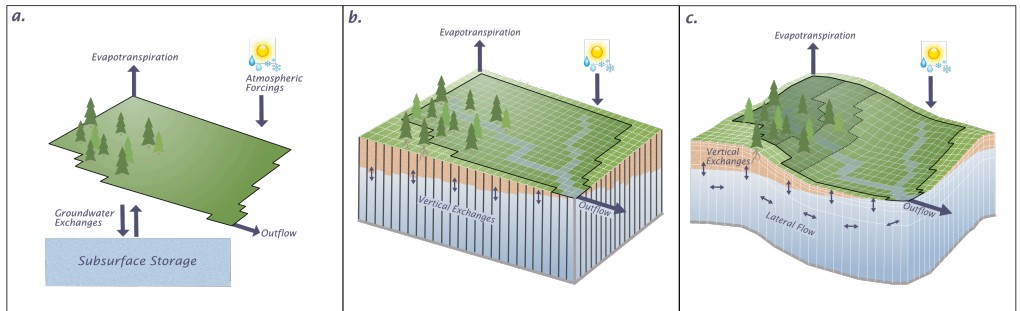


**Figure 1:** Conceptual illustration of (a) lumped parameter hydrologic models, (b) land
surface models with vertical subsurface exchanges and (c) integrated hydrologic models.
The nested subbasin approach is also illustrated on subplot c using the black outlines for
reference.


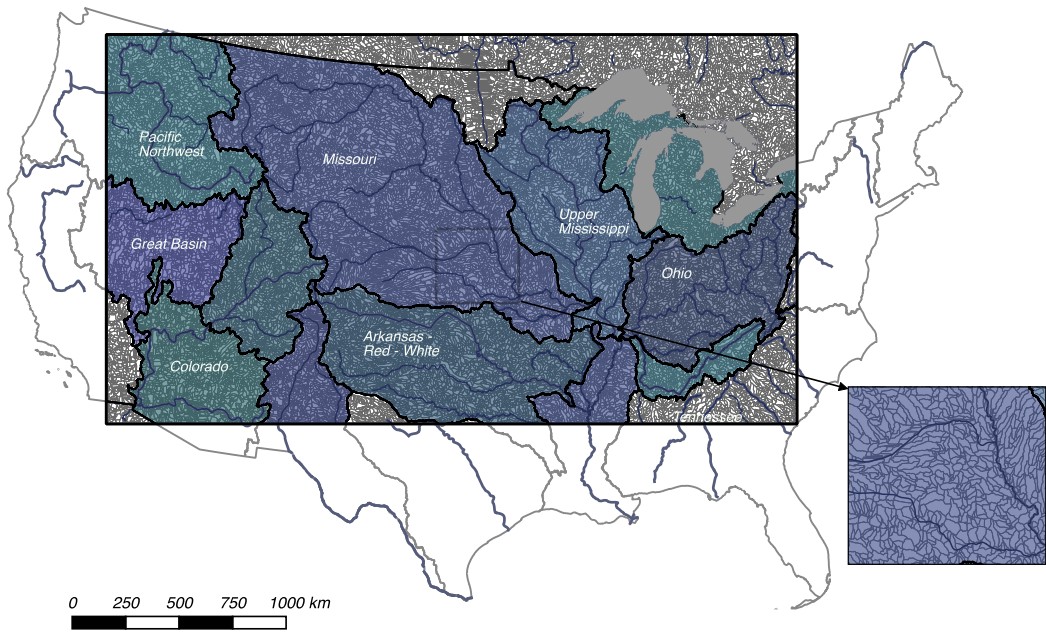


**Figure 2**: Map of the simulation domain extent (black box) with major river basins
highlighted and labeled. Subbasins within the domain are outlined in grey. Major rivers
are show in blue for reference (Note that the simulated river network is much more
highly resolved as illustrated in [*Maxwell et al.*, 2015])


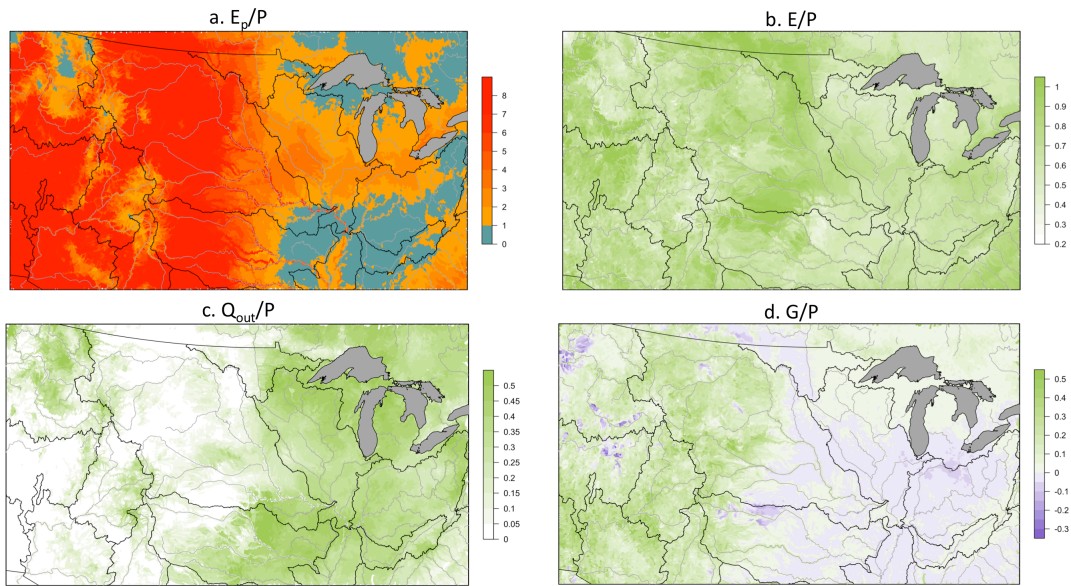


**Figure 3:** Maps of (a) aridity index (Ep/P) and the ratios of (b) evapotranspiration (c)

outflow and (d) groundwater contributions (G/P) compared to precipitation. Major river

basins are outlined in black.  Note that ratios are mapped according to the subbasins

shown in Fig. 2 but the values reflect the water balance for the entire watershed. This is

a system of nested watersheds so the value for each watershed is reported at its outlet

subbasin.

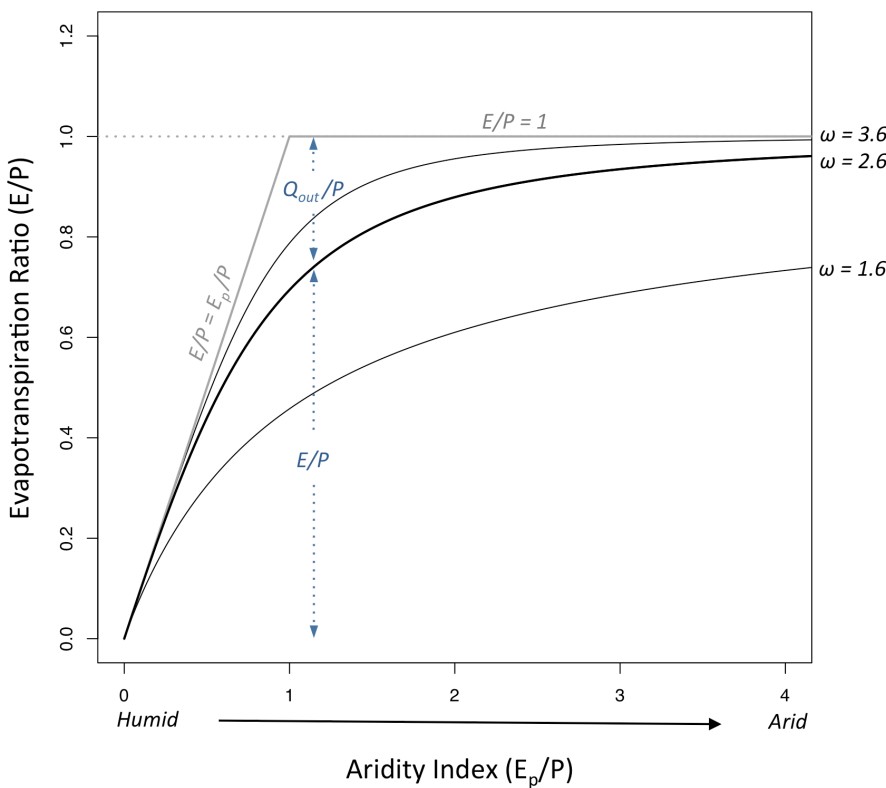

Figure 4: Illustration of the Budkyo framework showing curves with three different shape parameters (black lines, ω=1.6, 2.6 and 3.6) in relation to the water (E/P=1) and energy (E/P=$E_p$/P) limits of the system, grey lines.


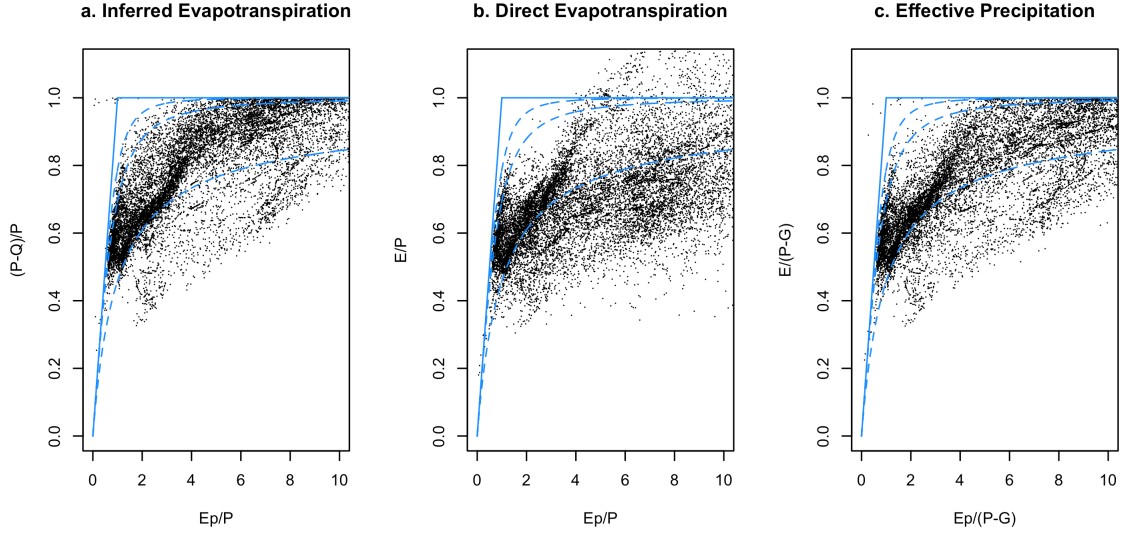


**Figure 5:** Budyko plots for the three approaches (a) inferred evapotranspiration, (b)
direct evapotranspiration and (c) effective precipitation with points for every watershed
in the domain. Dashed blue lines are Budkyko curves with ω values of 1.6, 2.6 and 3.6
and the solid blue lines are the water and energy limits (refer to Fig. 4).

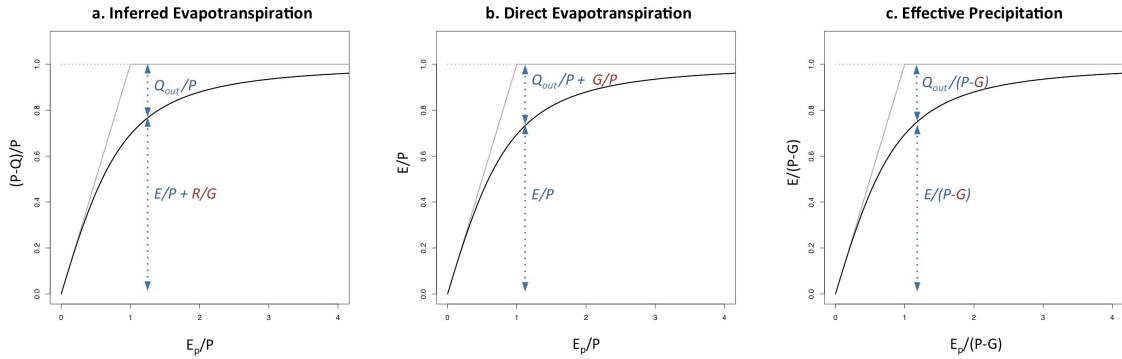

**Figure 6:** Illustration of the treatment of groundwater contributions for each of the
three approaches. The black lines show the water and energy limits and an example
Budyko curve, similar to Fig. 4.  Arrows indicate the water balance component
represented above and below the curve in each case.

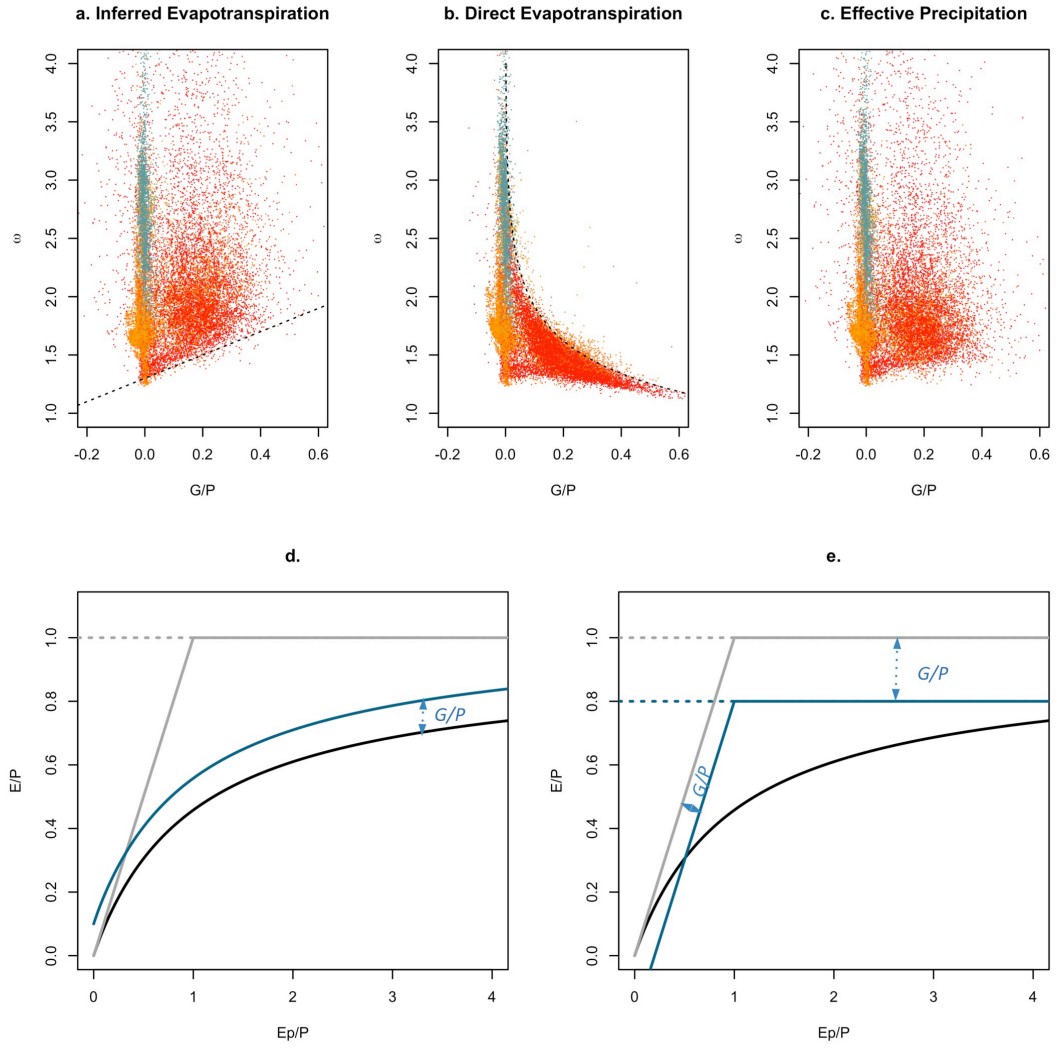


**Figure 7:** Comparison of shape parameters to groundwater contribution ratios for the three approaches in every watershed (a-c). Points are colored by aridity as shown in Fig. 3a. A dashed line with a slope of one is included on (a) for reference. The dashed line on (b) shows the relationship between the shape parameter and groundwater contribution fraction for an example aridity value of six in the limiting case where outflow is zero. The conceptual figures below illustrate the impact of a positive groundwater contribution (i.e. a net flux from the surface to the subsurface) for (d) the inferred evapotranspiration and (e) direct evapotranspiration approaches.

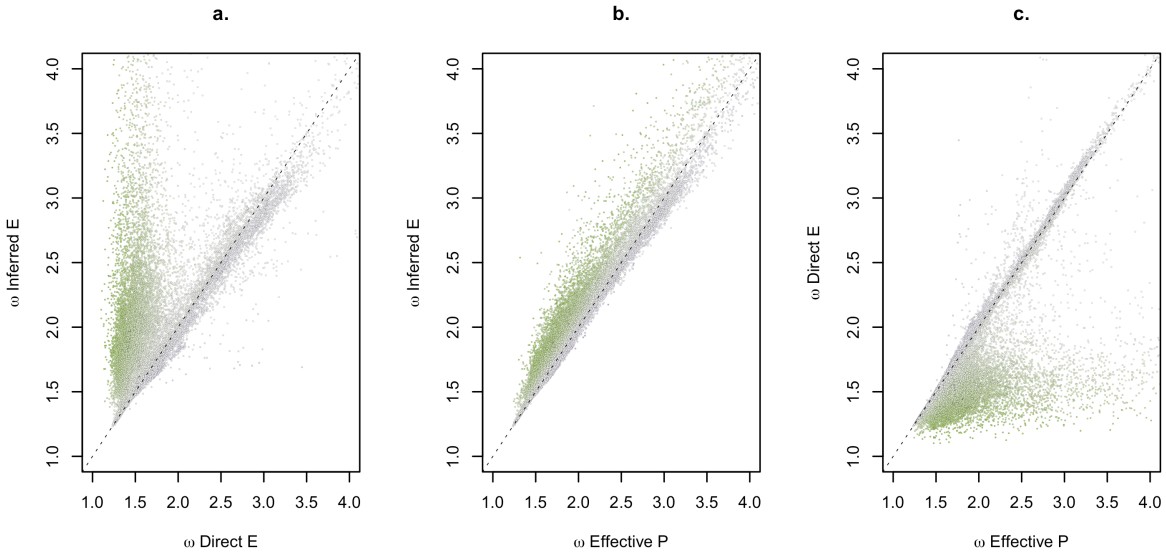


**Figure 8:** Comparison of shape parameters between the three approaches for every

watershed. Points are colored by groundwater contribution fraction as shown in Fig. 3d.

The dashed line on each plot is a one to one line for reference.

a. Inferred Evapotranspiration

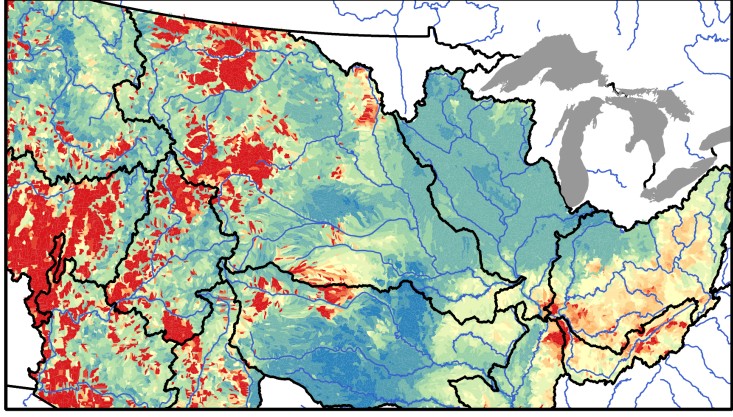

b. Direct Evapotranspiration

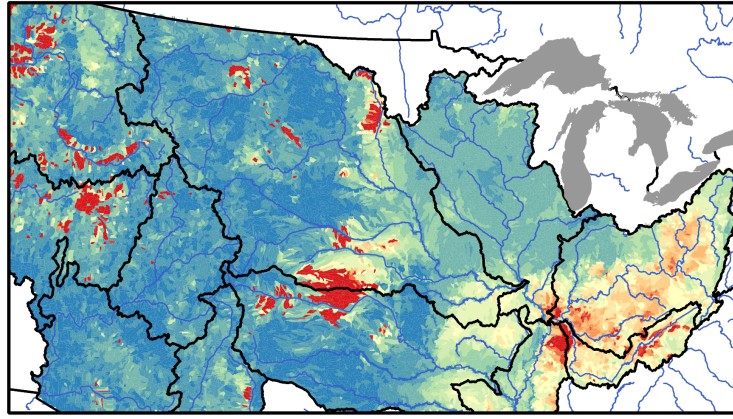

c. Effective Precipitation

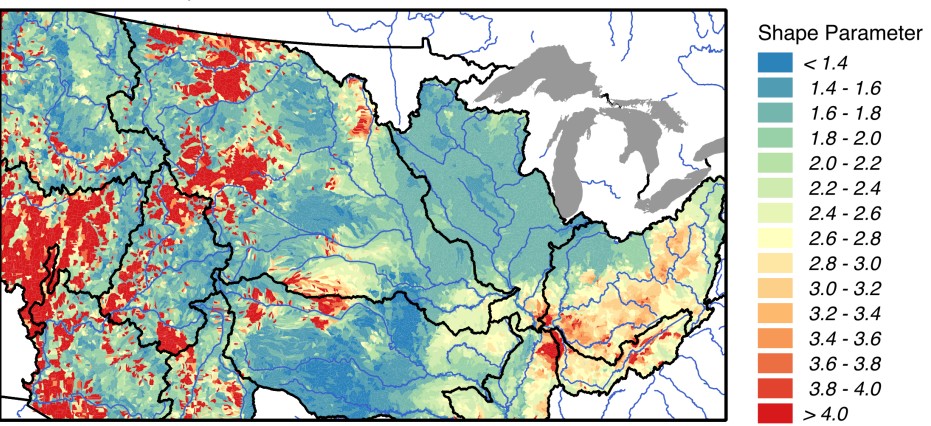

Shape Parameter
- < 1.4
- 1.4 - 1.6
- 1.6 - 1.8
- 1.8 - 2.0
- 2.0 - 2.2
- 2.2 - 2.4
- 2.4 - 2.6
- 2.6 - 2.8
- 2.8 - 3.0
- 3.0 - 3.2
- 3.2 - 3.4
- 3.4 - 3.6
- 3.6 - 3.8
- 3.8 - 4.0
- > 4.0


**Figure 9:** Map of shape parameters calculated for the 24,235 nested watersheds using
the (a) inferred evapotranspiration, (b) direct evapotranspiration and (c) effective
precipitation.  Major rivers are outlined in blue and regional boundaries in black.

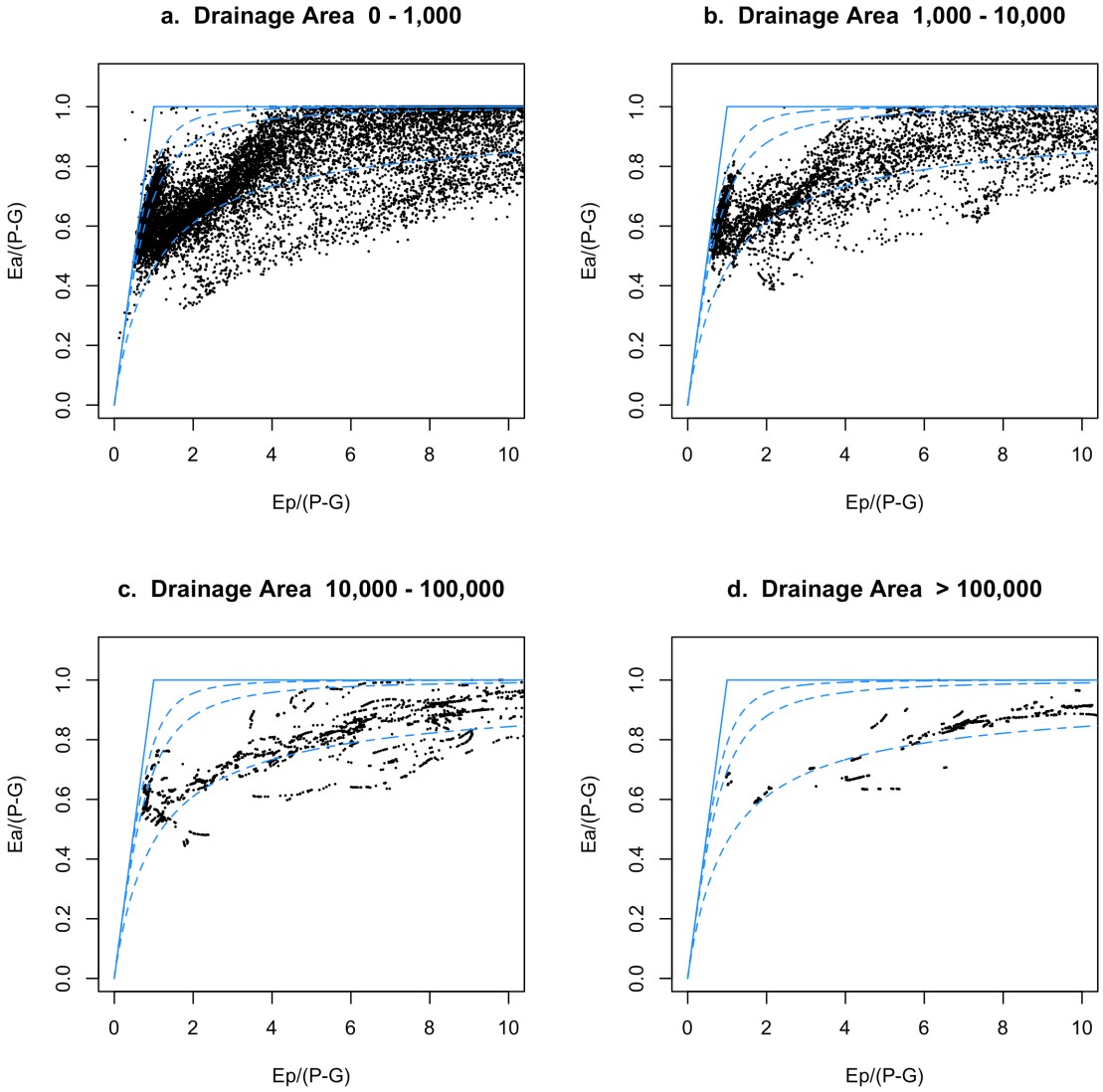


**Figure 10:** Budyko plots of evapotranspiration ratio versus aridity index using the

effective precipitation method with watersheds grouped by drainage area [km$^2$]. Blue

dashed lines are Budkyo curves with shape parameters of 1.6, 2.6 and 3.6 (refer to Fig.

4) and the solid blue lines show the water and energy limits.

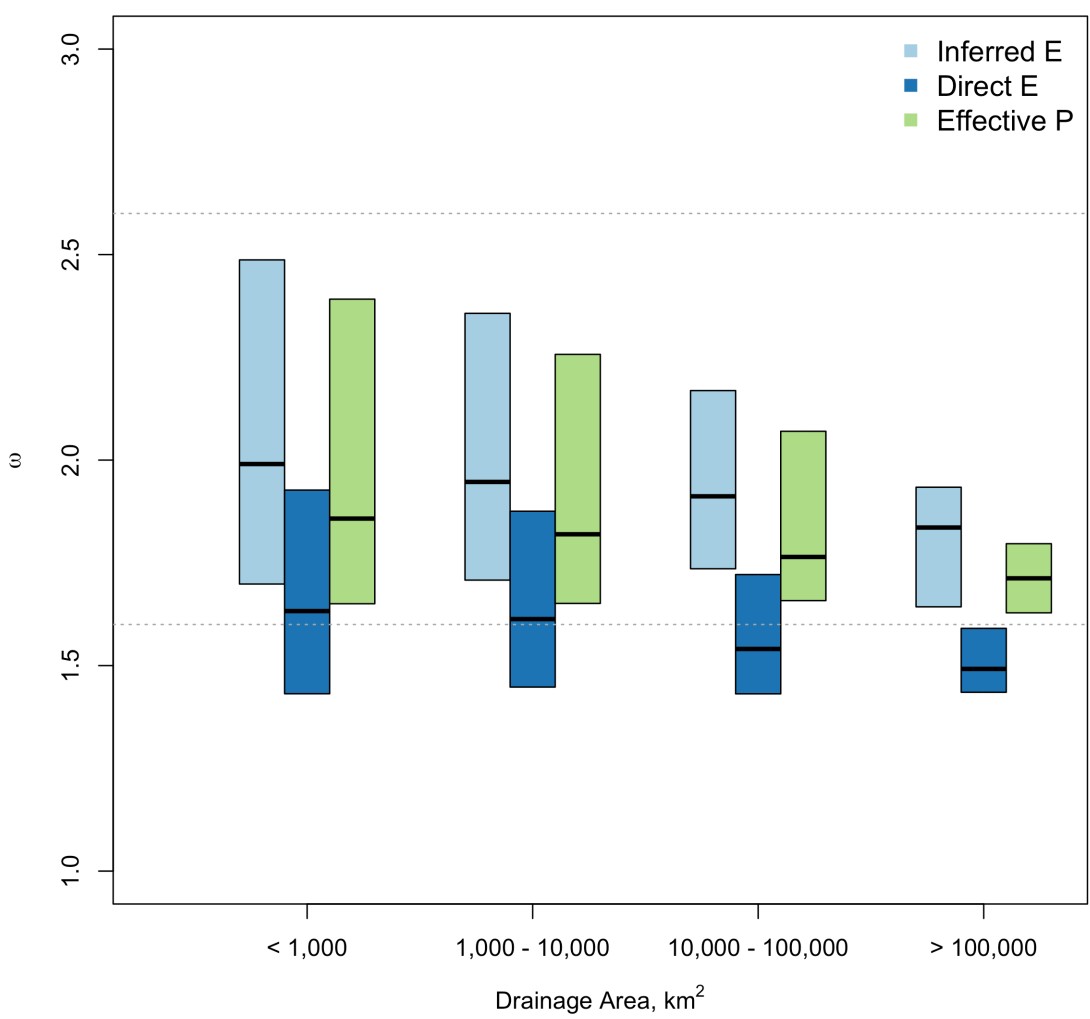

1174

**Figure 11:** Boxplots showing the interquartile range (i.e. 25-75[th] percentile values) of

shape parameters for all three approaches grouped by drainage area.  Dashed lines are

at 1.6 and 2.6 for reference.

1178