# Peer review of "Systematic shifts in Budyko relationships caused by groundwater storage changes"

_Hydrology and Earth System Sciences, 2016_

## Referee Comment (RC1) · Anonymous Referee #1 · 26 Sep 2016

My major concern on this MS are as follows:

The conclusions of this study are heavily relied on model performance: the validation of ET, groundwater, streamflow are needed, although the authors directed readers towards other papers.

The judgements are too strong: one-year simulation data was used to judge long term assumption; at least the authors should mention that they only check shortcoming of the short-term application of budyko hypothesis; all of the words in abstract and conclusions should be constrained on this aspect (short term application).

In the abstract, "trans-watershed lateral flow" (line 15) was mentioned, but only "groundwater surface water exchanges" are considered in this study as described in Methods section (2.4). If there exists "trans-watershed lateral flow", all the three methods should take it into consideration before comparison

In addition, some text wring skills also need more efforts: The descriptions of the methods should all put into the Methods section: e.g. lines 412-414; lines 415-436; lines 481-484

Small errors: Line 13: "sized"? Line 342: "than"?

---

## Referee Comment (RC2) · Anonymous Referee #2 · 28 Sep 2016

Overall, I find this manuscript to be interesting and in general, well written. It is also impressive to see these types of large, physically vigorous integrated GW-SW models applied towards problems that have typically been addressed with predominantly empirical methodologies. That being said, there are a number of issues with this work that need to be addressed prior to publication. Especially, I believe the authors need to address the major limitations in their approach.

Issue 1: Representation of the groundwater flow system.

As I understand from the description of the model domain, groundwater flow is primarily constrained within a single layer that extends from 2 m to 102 m below ground surface. Conceptually, this must mean that groundwater movement is restricted to a 2-

dimensional plane parallel to surface. With such simplification, can the model really be used to assess the groundwater component of the water balance within nested watersheds that range from 10's to 100,000's of sq-km? If one can assume the presence of Tothian flow systems across much of the domain, how can local groundwater flow conditions develop when in essence the local systems will be overprinted by large regional flow? If the results of the work did not have such strong dependence on the model's ability to simulate the water balance in small watersheds this would not be such an issue; however, given the large number of small watersheds that are considered in the analysis, I believe it is crucial to have a more highly resolved subsurface domain. Even one or two additional layers within the groundwater flow zone would allow 3 dimensional groundwater flow systems to develop, and hence facilitate the model's ability to capture local flow systems that are a key part of the water balance, especially in humid areas with notable topographic variability such as the Eastern and Northwestern regions in the domain. Furthermore, a graphic that depicts how the watershed nesting has been conceptualized within the model domain framework would be of value to readers.

Issue 2: Applicability of the Gleeson et al. (2011) dataset.

While it does need to be recognized that parameterizing the subsurface component of these large integrated models is challenging, and the use of large scale homogeneous datasets is particularly attractive, modellers must be cognizant of the limitations that these datasets impose on the model. This point is highlighted with the Gleeson et al. (2011) subsurface permeability map, which is an extreme simplification of subsurface hydrostratigraphy in order to facilitate global coverage. For an application such as the one of focus here, the Gleeson et al. dataset does not provide adequate spatial accuracy/resolution for credible model results to be generated for smaller-scale watersheds, and considering the large weighting that smaller-scale model results for the groundwater component of the water balance are given in the analysis, this is very problematic.

Issue 3: The use of modelled ET.

Typically, and especially in large scale models, it very difficult to accurately simulate ET. In this work the authors use simulated ET as a surrogate for measured ET across much of continental USA. Considering the extreme importance of ET in the water balance analysis, I wonder if biases or errors in simulated ET may not be skewing the results. This point may be highlighted in Figure 2d, where it appears that groundwater recharge is unrealistically high across much of the Great Plains. Furthermore, as Figure 10 highlights, the analysis results that are dependent on simulated ET show a strong deviation from the results generated from the other two water balance calculation methodologies.

A few other minor comments are as follows:

L16: be careful with use of 'realistic' this work is more conceptual in nature

L96: . . .a physically. . .

L120: abcd?

L135: expense,

L142: comma not needed

L150: technical feasible yes, but how realistic is it to extract local scale information from continental scale models?

Eqn 1: Are the units expected to balance?

L167: Verify units for q

L201: Ev, and. . .

L263: Is a single year really ideal?

General comment: a histogram showing watershed size distribution would be valuable.

L298: balanced

L314: for

L331: opposed to

L340-343: This statement is not really supported by Figure 2, which is presumably (authors should state this in caption) depicting ratios in annual totals. This statement is also irrelevant to main objectives, suggest deletion.

L396: one,

L410: and often

L437 onwards: these results and discussion should be supported by at least some basic curve fit statistics.

L514: correlation discussion should be supported by some r^2 values.

L576: higher curve numbers? Visually the results look the same, could statistics be used here as well?

L578: do you mean Fig. 6?

L603: 100? Do you not mean 1000?

L607: References needed here.

Figure 9: General observation here: As the authors state, the analysis results that focus on the larger watersheds provides a better match to the idealized curves. Could this result be at least in part due to the major issues identified earlier in this review that highlight the weaknesses in the authors approach towards simulating the groundwater component of the water balance for small watersheds?

L612: No doubt watershed characteristics are important, but what about limitations with the way groundwater flow in small watersheds is represented in the model, and the adequacy of the employed datasets for meeting such finely resolved objectives?

Summary and Conclusions: General comment for this section is that it is too long and

there is too much detail provided in the reiteration of methodology and results. Suggest shortening by at least 50% and focusing more on the significance of the results.

---

## Referee Comment (RC3) · Anonymous Referee #3 · 9 Oct 2016

Overall, I think this paper has the potential to turn into a good contribution that elaborates the influence of groundwater on the Budyko Hypothesis. The paper does not seem to have a well-described objective. I did not see a set of research questions or hypotheses to be tested. All the results presented in the paper are based on a single water year simulation in the ParFlow model, which is a fairly short time scale to convincingly report and use any groundwater related modeled variables. As I tried to figure out what the objectives of this paper might be I kept asking myself the following questions. Is the idea to:

1) develop a conceptual model for incorporating the role of groundwater (GW) to the Budyko hypotheisis (BH)?

2) parameterize the contribution of GW in the BH by relating the w parameter in the Zhang equation to a GW variable that may be obtained from observations or models, which can be used as a simple model?

3) evaluate model results to see when a Budyko type behavior is generated in systems where GW cannot be neglected (e.g. Fig 4), by modifying the source of water in the axis of the BH plotting position?

It was never clear to the reader why three water balance conceptualizations were used and why w were calculated for all three of them (Figs 5, 6, 7). The authors need to state what their goals were.

If one needs to improve the use of the BH for regions where GW can not be neglected, one could work with the original model inputs of observed P and Q, and calculated Ep, and parameterize w = f(GW, E/P, Ep/P) and use this w in the original model and test it .. In your case E would come from PArFLow. Apparently this does not seemed to be the objective of this paper, but I felt that Fig 6b came close to this idea but stopped there.. Finding w value for the indirect method (Fig 6a) did not make sense to me as E=P-Q won't give the "correct" ET and therefore why would you calculate w using this ET/P. Please better state what you objectives are.

Interpretation of Figure 4,5,6,7 need help. The paper does not sufficiently discuss the processes that lead to patterns in these figures.

Abstracts lines 25, 26—what do you mean by best results? Best of what? Is the "best" represent better predicted water balance by the BH, modified in this study, against modeled water balance? Or did you develop a simpler model of water balance that gives consistent predictions with ParFlow?

Modeling methods: In this study modeled data comes from ParFlow, which was used for only a single water year (1985), starting from a steady-state groundwater configuration. Obviously the question is – why would you use a single water year.. I wonder if

the system won't respond to this steady-state assumption when you start running the model with the actual climate forcing of 1985. The paper mentions that PArFlow simulations were done for historical climate in CONUS. I wonder why the authors did not use the full length of simulations and evaluate the –mean annual water balance with GW contribution in the BH hypothesis, instead of just using a single year which I presume creates some rapid transient conditions in the beginning of the model run as the water table would respond to the 1985 forcing. Running the model with a historical climate forcing data and evaluating the long-term water balance with long-term-average estimated flux variables, including groundwater seems to be the logical way to go. I'm having a hard time accepting the justification of the use of a single water year. BH is ideal for long-term-aver water balance conditions as well. So logic tells me to use longer simulations.

In the methods the G term need to be more clearly explained in my opinion. In reading the paper I went back and forth a few times to make sense of what authors might have meant by G but I'm still not clear.. My intuition tells me that groundwater contribution would be the net volume of groundwater staying in the basin at the end of a water year.. I imagine G is not always a contribution as in some cases G may flow out of a basin in which case G will be a sink term. Your groundwater surface water exchange can practically be infiltration or saturation excess overland flow.. I'm not following what this definition means in the context of eq.(6). Your sign convention in the G/P plots should be explained.

Line 319– I'm not following this para.. shouldn't a positive G mean that the watershed receives flux across its groundwater boundaries and a negative G indicates net export of GW to surrounding basins.. Water that infiltrates to subsurface would just increase the storage of GW wouldn't it.. This water may stay in the watershed or exported out.. seems like concepts are a bit miss-use here or not explained clearly.. Perhaps you use a Delta Storage term in 6 and 7 and explain these referring to the storage change etc.. hard to follow here..

Line 353- I'm not clear how G was calculated.. Above you said you used eq (6).. here G/P>0 is interpreted as storage gain.. Headwater of Missouri should be recharging the system and therefore they are not GW exporters.. but the region below in ND and Nebraska area should be net exporters right..? so I was expecting to so G/P<0 in northern basins and G/P>0 toward the middle of Missouri where it connects to Mississippi.. Please better explain conceptual model. Please clarify– if G>0, there should be net input to the watershed and Effective precip should be P+G, and if G<0 there is net export from the watershed and Effective P=P-G... Is your formulation consistent with this?

Line 419.. I would not cite Istanbulluoglu et al., 2012 here. Istanbulluoglu et al showed the limitation of assuming ET=P-Q in the Budyko curve, and proposed to use ET= P-Q-DeltaS, where DeltaS is change in groundwater storage assuming no net export/import of GW.

Line 453..don't cite Istanbulluoglu et al., 2012 here

Line 525.. Istanbulluoglu et al., 2012 used the inferred ET approach to show its limitations– not as the proposed method to calculate ET from P and Q. Incorporating the contribution of groundwater in the water balance equation to calculate ET led to a more consistent trend in the evapotranspiration ratio and aridity index.. See Figs 6a,b and Fig 7a,b,e,f. I think this paragraph should better summarize their results.

---

## Referee Comment (RC4) · Anonymous Referee #4 · 21 Oct 2016

The authors have used a physically based hydrological model to improve water-budgeting at catchment scale. In particular, they have considered the original Budyko model as a reference and shown that by accounting for ground water inflow/outflow, water budgeting can be done more accurately. I really appreciate the authors' effort to undertake such an extensive numerical analysis. The article looks suitable for publication although I think a couple of key concerns the authors need to address.

1. Purpose of the study

The authors need to elaborate on the usefulness of their study. The physically based hydrological model they are using has many parameters; they cannot take that model to a random ungauged catchment and predict its hydrological variables. On the other

hand, the Budyko model is a universal deterministic model which can be applied to any ungauged catchment. It is thus not surprising that the multi-parameter model will perform better after calibration. I don't think their study is very informative unless they integrate a deterministic physically based hydrological model with the Budyko model to improve prediction.

2. Clarity of presentation.

It is quite hard to follow what the authors are saying at many places. In my opinion, the presentation needs to be simple. If the authors' objective is to show how the physically based hydrological model is doing a better job at water budgeting, they need to focus on that part more. There is not a single figure showing a direct comparison between prediction by the physically based hydrological model and that by the Budyko model... Terms need to be defined prior to their usage. For example, in Line 27 the authors are talking about Budyko curve parameters. The authors are actually talking about Fu model's parameters (Budyko model does not have any parameter).

---

## Author Comment (AC1) · 18 Nov 2016

**We thank the referee for their constructive comments. We have provided detailed responses to each of their points below (author response is in bold).**

My major concern on this MS are as follows:

- The conclusions of this study are heavily relied on model performance: the validation of ET, groundwater, streamflow are needed, although the authors directed readers towards other papers.

   **The purpose of this study is to evaluate the influence of groundwater surface water exchanges on the shape of Budyko relationships given different approaches to quantifying evapotranspiration. We agree with the reviewer that the accuracy of local shape parameters for water year 1985 will be heavily dependent on the performance of the model. However, our goal here is not to predict shape parameters for our one-year simulation. Rather, we are using the model as a numerical testbed where we have fully defined precipitation, evapotranspiration, storage changes and runoff to evaluate the influence of different accounting methods on the resulting behavior. Thus, how well the model performs compared to observations is much less of a critical path toward the conclusions of this current work. Still, the model has been exhaustively validated to more than 1.2M observations [*Maxwell and Condon*, 2016]. As this appears in a 12,000-word, 20 figure document we feel directing the reader there is a much better approach than trying to replicate the validation in the current work. Of course the reviewer is correct that a better fit between the simulated results and observations for water year 1985 would improve our ability to use the Budyko relationships developed here to predict watershed evapotranspiration, but it would not change the findings of this paper.**

   **In response to the rest of the comments from this referee, as well as the other referees, we have significantly modified the manuscript to (1) more clearly emphasize the purpose of this analysis and (2) better explain our use of modeling as a numerical testbed and not a predictive framework. We hope that with these changes will more clearly emphasize the fact that the results presented here are general relationships that are not reliant on the underlying model which was only used as a means to sample the Budyko space within a controlled numerical framework.**

- The judgements are too strong: one-year simulation data was used to judge long term assumption; at least the authors should mention that they only check shortcoming of the short-term application of budyko hypothesis; all of the words in abstract and conclusions should be constrained on this aspect (short term application).

   **We agree with the referee that the one-year simulation used here does not prove or disprove whether it is appropriate to assume equilibrium conditions for long-term simulations. Our intention is not to predict when and where this assumption is valid, rather we seek to investigate the impact of storage changes when they are occurring. We intentionally chose to evaluate a one-year simulation because it is not in dynamic equilibrium and therefore captured a range of storage changes across the simulation domain. This allowed us to demonstrate the impact of variable groundwater storage changes on Budyko relationships. The intent here is only to demonstrate potential impacts given different approaches to evapotranspiration for a range of storage changes, this is not dependent on the time frame these changes occur over.**

   **As stated prior, we have refocused the purpose of our analysis and its intended applications in the revised manuscript in response to the reviewer comments. We have modified the introduction**

**and methods significantly to better clarify our goals and to be more explicit about the reason we selected a one-year simulation. Additionally, the conclusions and abstract have been refocused around these points to better highlight the intended use of these results.**

- In the abstract, "trans-watershed lateral flow" (line 15) was mentioned, but only "groundwater surface water exchanges" are considered in this study as described in Methods section (2.4). If there exists "trans-watershed lateral flow", all the three methods should take it into consideration before comparison

**In response to other comments the abstract has been significantly revised to better reflect the purpose of this work. In the revised abstract we do not use this term. Additionally, we would like to clarify that the focus here is on the net exchange between groundwater and surface water so lateral groundwater fluxes are not explicitly analyzed for this work. However, they are simulated in the model and are an important drivers of exchanges. We have added the following text to the methods section to be more explicit about this point:**

> *"This approach is focused solely on the net contribution of groundwater to the surface water budget. Nested systems of local and regional lateral groundwater flow are simulated within the model and previous work has evaluated spatial patterns and physical drivers of lateral groundwater imports and exports across the domain [Condon et al., 2015; Maxwell et al., 2015] as well as groundwater residence times [Maxwell et al., 2016]. Here we focus only on net exchanges with the surface that are relevant to the Budyko formulation. We do not need quantify lateral exchanges in the subsurface directly for these purposes; however, it should be noted that the lateral redistribution of groundwater that occurs within the model is still vital to generating realistic groundwater configurations and supporting groundwater surface water exchanges."*

- In addition, some text wring skills also need more efforts: The descriptions of the methods should all put into the Methods section: e.g. lines 412-414; lines 415-436; lines 481-484

**In response to this comment we created an additional sub-section in the methods that covers the three approaches to evapotranspiration calculations and we moved the description of Budyko calculations on lines 481-484 into the methods section on Budyko analysis (Section 2.5).**

- Small errors: Line 13: "sized"? Line 342: "than"?

**Thank you, these have been corrected in the revised manuscript.**

**Works Cited:**

Condon, L. E., A. S. Hering, and R. M. Maxwell (2015), Quantitative assessment of groundwater controls across major US river basins using a multi-model regression algorithm, Advances in Water Resources, 82, 106-123, doi: http://dx.doi.org/10.1016/j.advwatres.2015.04.008.

Maxwell, R. M., and L. E. Condon (2016), Connections between groundwater flow and transpiration partitioning, Science, 353(6297), 377-380, doi: DOI: 10.1126/science.aaf7891.

Maxwell, R. M., L. E. Condon, and S. J. Kollet (2015), A high resolution simulation of groundwater and surface water over most of the continental US with the integrated hydrologic model ParFlow v3, Geoscientific Model Development, 8, 923-937, doi: 10.5194/gmd-8-1-2015.

Maxwell, R. M., L. E. Condon, S. J. Kollet, K. Maher, R. Haggerty, and M. M. Forrester (2016), The imprint of climate and geology on the residence times of groundwater, Geophysical Research Letters, 43(2), 701-708, doi: 10.1002/2015GL066916.

---

## Author Comment (AC2) · 18 Nov 2016

**We have provided point by point responses below with author response is in bold.**

Overall, I find this manuscript to be interesting and in general, well written. It is also impressive to see these types of large, physically vigorous integrated GW-SW models applied towards problems that have typically been addressed with predominantly empirical methodologies. That being said, there are a number of issues with this work that need to be addressed prior to publication. Especially, I believe the authors need to address the major limitations in their approach.

We thank the referee for their thoughtful review of our manuscript. We completely agree that, as with other approaches, there is no perfect solution and there are still many limitations to this type of modeling. We hope that the reviewer will see that we have kept these limitations in mind when designing our experiment and in determining what type of questions these simulations are well suited to answer. We have clarified these points in our responses below and have also made significant revisions to manuscript to more clearly explain the purpose of our comparative analysis as well as its limitations.

Issue 1: Representation of the groundwater flow system.

As I understand from the description of the model domain, groundwater flow is primarily constrained within a single layer that extends from 2 m to 102 m below ground surface. Conceptually, this must mean that groundwater movement is restricted to a 2- dimensional plane parallel to surface. With such simplification, can the model really be used to assess the groundwater component of the water balance within nested watersheds that range from 10's to 100,000's of sq-km? If one can assume the presence of Tothian flow systems across much of the domain, how can local groundwater flow conditions develop when in essence the local systems will be overprinted by large regional flow? If the results of the work did not have such strong dependence on the model's ability to simulate the water balance in small watersheds this would not be such an issue; however, given the large number of small watersheds that are considered in the analysis, I believe it is crucial to have a more highly resolved subsurface domain. Even one or two additional layers within the groundwater flow zone would allow 3 dimensional groundwater flow systems to develop, and hence facilitate the model's ability to capture local flow systems that are a key part of the water balance, especially in humid areas with notable topographic variability such as the Eastern and Northwestern regions in the domain. Furthermore, a graphic that depicts how the watershed nesting has been conceptualized within the model domain framework would be of value to readers.

The referee is correct that we have a simplified representation of the subsurface and we agree that additional layering and increased resolution would improve the simulation. However, we would like to clarify that the current approach does not mean that all groundwater flow is occurring parallel to the surface. Previous work has evaluated groundwater flow systems within this model and showed power law residence time distributions as well as nested systems of local and regional groundwater convergence [*Condon and Maxwell*, 2015; *Condon et al.*, 2015; *Maxwell et al.*, 2016]. We do observe systems of local convergence and our ability to balance water does not depend on the watershed size. ParFlow is simulating a gridded domain and it balances water for every grid cell and time step. The nested watersheds are only used to group results for analysis as a post processing step and are not units of simulation. To make this point more clearly we have added the following conceptual figure to the methods section which compares our integrated hydrologic model (subplot c) to lumped parameter models (subplot a) and gridded models which incorporate only vertical interactions with the subsurface (subplot b). Subplot c of this figure also illustrates the nested watershed approach used here.

Also, we would like to stress that the purpose of our analysis is not to predict the magnitude of storage changes across the US for water year 1985. Rather, we are using this simulation as a way to sample a broad range of groundwater storage changes in order to illustrate the way these changes would impact various Budyko approaches. We agree that improving the subsurface characterization could change the level of storage changes that we simulate, however these changes would not impact our findings. We do not think that this point was made clearly in the original manuscript and we have significantly revised the text to better clarify our goals.

Issue 2: Applicability of the Gleeson et al. (2011) dataset.

While it does need to be recognized that parameterizing the subsurface component of these large integrated models is challenging, and the use of large scale homogeneous datasets is particularly attractive, modellers must be cognizant of the limitations that these datasets impose on the model. This point is highlighted with the Gleeson et al. (2011) subsurface permeability map, which is an extreme simplification of subsurface hydrostratigraphy in order to facilitate global coverage. For an application such as the one of focus here, the Gleeson et al. dataset does not provide adequate spatial accuracy/resolution for credible model results to be generated for smaller-scale watersheds, and considering the large weighting that smaller-scale model results for the groundwater component of the water balance are given in the analysis, this is very problematic.

We agree with the referee that subsurface datasets remain a significant limitation for large scale groundwater simulations. The permeability map developed by Gleeson is not perfect, but it is the only consistent dataset available for the entire continental US (it should be noted also that the resolution of this dataset is higher for the US than it is globally). If the goal is to build a model to precisely predict local groundwater levels in small scale systems, we agree with the referee that this is not the ideal dataset to start from. However, this is not the intention of this work. We focus on using this model as a tool to characterize behaviors across a broad range of physical settings and spatial scales that incorporate realistic heterogeneity. We have intentionally designed our experiment so that the credibility of our results is dependent on the physical processes we are simulating and not any local calibration. We have added the following text to the methodology section to make this point more clearly to other readers (new text is underlined):

"The 1-year simulation presented here intentionally violates the steady state assumption. The purpose of our analysis is to evaluate the impact of net storage changes on Budyko relationships, therefore a steady-state simulation is not the goal. It can also be argued that storage changes will vary from year to year or depending on the multi-year period analyzed. The 1985 simulation year is not presented as a prediction of long-term storage variability, it is simply used to sample a range of groundwater surface water exchange across variable climates and physical settings. We present a general framework for understanding the impacts of storage changes in various Budyko formulations using water year 1985 as a representative example.

Similarly, because we are focused on a comparative analysis within the Budyko framework the results are not dependent on local calibration between simulated results and observations. The discrepancies between approaches stem from differences in the variables used to create a water balance (refer to sections 2.3 and 2.4); these findings are not sensitive to parameter uncertainty in the model. Still, the transient simulation has been rigorously validated against all publically available observations for water year 1985. This includes transient observations at varying frequencies from 3,050 stream gauges, 29,385 groundwater wells and 378 snow stations for a total of roughly 1.2 million comparisons points. Flux tower observations were not available over this period, but latent heat fluxes were also compared to the Modern Era Retrospective-analysis for Research and Application (MERRA) dataset. Complete details of the model validation are provided in the supplemental information of Maxwell and Condon [2016].

Although there are of course limitations to the model and significant uncertainties in spatial model parameterization, especially for the subsurface, overall comparisons between simulated and observed values demonstrate that the modeling approach is robust. Streamflow timing and magnitude are generally well matched in undeveloped basins, snowpack timing and melt is accurate and spatial patterns in latent heat flux are reasonable. Most importantly for this analysis, the model validation shows that ParFlow is accurately capturing the relevant physical processes. Uncertainty in subsurface parameterization, bias in atmospheric forcing data and lack of anthropogenic activities were identified as key areas that could improve the local predictions of the model. However, as discussed above, the purpose of this work is not to predict Budyko curve parameters for water year 1985. The uncertainties listed here are therefore important to note, but do not limit the utility of this tool as a test bed for evaluating interactions across spatial scales and complex physical settings."

Issue 3: The use of modelled ET.

Typically, and especially in large scale models, it very difficult to accurately simulate ET. In this work the authors use simulated ET as a surrogate for measured ET across much of continental USA. Considering the extreme importance of ET in the water balance analysis, I wonder if biases or errors in simulated ET may not be skewing the results. This point may be highlighted in Figure 2d, where it appears that groundwater recharge is unrealistically high across much of the Great Plains. Furthermore, as Figure 10 highlights, the analysis results that are dependent on simulated ET show a strong deviation from the results generated from the other two water balance calculation methodologies. All of our analysis is based on simulated results because our goal is to characterize the way that storage changes perturb points within the Budyko space. We have attempted to clarify this point

more fully in the following text that was added to the start of the methods section:

"We use an integrated hydrologic model to simulate water and energy fluxes in both the surface and the subsurface. Here we apply a high resolution (1 km2) simulation of the majority of the continental U.S. which covers more than 6 M km2 and simulates hydrologic systems across a broad range of physical settings and storage change magnitudes. The model is driven using historical observed atmospheric forcings such as precipitation and temperature, and provides gridded outputs of all water and energy fluxes throughout the system. We use simulated surface water flow, evapotranspiration and groundwater surface water exchanges to calculate Budyko relationships using three different approaches to estimate fluxes:

- 1. Calculating evapotranspiration from simulated runoff and precipitation
- 2. Using simulated evapotranspiration values directly

**3.** Using simulated evapotranspiration values directly and taking into account storage changes.

Differences between the approaches are compared with storage changes in each basin to evaluate the systematic impacts of these changes on Budyko relationships.

The numerical modeling approach used here provides several important advantages for this type of analysis. Within the numerical framework, groundwater surface water exchanges for every watershed in the system are fully characterized. This guarantees perfect closure of the water balance and means that we can mimic all three approaches within a consistent numerical framework where storage changes are directly accounted for. Furthermore, because the goal is to understand differences between approaches, and not to predict local Budyko parameters, the key advantage here is the ability to evaluate physically realistic behavior across a variety of physical settings and spatial scales where groundwater can be fully accounted for. Within this context, it should also be noted that the focus is on how groundwater storage changes perturb relationships. Therefore, uncertainty in local model parameters is much less important than realistic simulation of physical interactions for a range of storage changes and aridity values within a controlled numerical framework."

Also, we have added the following note to the description of the direct evapotranspiration method to emphasize this point:

"Note that in this case we are still using simulated E not observations. The intention is to treat the model as our simulated truth and compare variations within this framework"

Figure 10 does show clear differences between the direct evapotranspiration approach and the other approaches, but this is not an indication of model bias because all of the results in this figure are based on the same simulated outputs. The point of figure 10 is to show that the way storage changes are accounted for in different methods will systematically impact results. In all cases, the data underlying the plots is exactly the same; we are just showing here that you get a different answer depending on what parts of the water balance you choose to analyze. Based on this comment and others we think that this point was not made clearly enough in the original manuscript. We have expanded the discussion of all of the results figures to try to emphasize these points better.

A few other minor comments are as follows:

L16: be careful with use of 'realistic' this work is more conceptual in nature We agree that 'realistic' is a relative term and in the revised abstract we do not include this language.

L96: . . .a physically. . .

This is no longer included in the revised manuscript

L120: abcd?

'abcd' is the name of the hydrologic model applied by Du et al.

L135: expense, This sentence has been revised.

L142: comma not needed This has been corrected in the revised manuscript.

L150: technical feasible yes, but how realistic is it to extract local scale information from continental scale models?

This is of course an open point of debate. We would argue that even if there is local uncertainty there is still much that can be learned about spatial organization and process interactions across scales from these models. Please refer to our responses to the major comments above for a more detailed response on model limitations.

Eqn 1: Are the units expected to balance?

Yes, we have confirmed that units of this equation do balance. The units of every term are 1/time.

**L167: Verify units for q**

The units for q are correct as stated (i.e. 1/time). Fluxes are normalized by the cell thickness in ParFlow so the length dimension is canceled out in this formulation. We have added the following text to the manuscript to clarify this point:

"Note that units of  $T^1$  for the flux terms reflects the fact that they are scaled by the cell thickness."

L201: Ev, and. . .

This has been corrected in the revised manuscript.

**L263: Is a single year really ideal?**

In this sentence we were intending to say that the pre-development approach was ideal. We have revised the sentence as follows:

"Therefore, the simulation represents natural flows in a pre-development scenario, which is ideal for Budyko analysis."

General comment: a histogram showing watershed size distribution would be valuable. We appreciate the suggestion. In response to earlier comments we decided to add an additional conceptual figure explaining our modeling approach and the system of nested watersheds we are using. We have decided not to add additional histogram figure here; however, we hope that the reviewer will see from this response that our findings are not sensitive to the size distribution.

L298: balanced This has been corrected.

L314: for This has been corrected.

**L331: opposed to This has been corrected.**

L340-343: This statement is not really supported by Figure 2, which is presumably (authors should state this in caption) depicting ratios in annual totals. This statement is also irrelevant to main objectives, suggest deletion.

The text on lines 340-343 explains that groundwater contributions fractions are significant throughout the domain and therefore the system is not in a steady state over the annual simulation period. We are a little confused by the comment that this statement is irrelevant to the main objectives of our work because the entire purpose of this analysis is to evaluate the impact of storage changes on Budyko relationships. It is in our opinion critical to show that we have selected a simulation period where storage changes are occurring. We have clarified in the caption of Fig. 2 that subplot d is plotting G/P and we have revised the sentences in question as follows to hopefully be more explicit about this point:

"Within this annual simulation, subplot d shows that groundwater surface water exchanges (G/P) can be a substantial portion of the water balance in much of the domain. This indicates that the system in not in steady state over the simulation period. As discussed in Section 2.2 the one-year simulation time was intentionally selected for this reason. Here, we take advantage of the ability to directly calculate groundwater surface water exchanges within a controlled numerical simulation where such exchanges are prevalent in order to evaluate the impact of storage changes on Budyko relationships across a range of spatial scales and climates."

L396: one, This coma has been added.

L410: and often

This text has been removed from the revised manuscript.

L437 onwards: these results and discussion should be supported by at least some basic curve fit statistics.

It is not possible to fit the results to a single curve because the premise here is that different points are falling on curves with different shape parameters. We agree that some quantitative metrics are needed though. In response to this comment we have evaluated the number of points falling within the bounds of the curves defined with shape parameters of 1.6 and 3.6 and revised the associated text as follows:

"Fig. 4 plots every watershed in the domain shown in Fig. 1 using the three approaches to estimate the evapotranspiration fraction. In all three figures, the watershed points follow the overlaid Budyko curves; 77% of the watersheds fall within the 1.6 to 3.6 shape parameter lines for the inferred evapotranspiration approach, 51 % for the direct approach and 72% for the effective precipitation approach."

L514: correlation discussion should be supported by some r2 values.

In this case it is not possible to calculate an r2 value because we are just talking about the relationships at the lower limit of the plot. However, we have revised the text in question as follows:

"Finally, a scatter plot of shape parameters versus groundwater contribution fraction for the effective precipitation case (Fig 6c) shows similar patterns with aridity but no clear correlation between storage changes and shape parameters for the lowest shape parameter values."

Additionally, we have added r2 values to the discussion of Figure 7 as follows:

"Also, there is a much stronger correlation between the inferred evapotranspiration and effective precipitation approaches (Fig. 7b) than between direct evapotranspiration and effective evapotranspiration approaches (Fig. 7c) (r2 value of 0.96 comparing inferred vs. effective as opposed to 0.32 for inferred vs. direct)."

L576: higher curve numbers? Visually the results look the same, could statistics be used here as well? In response to this comment we have expanded this discussion as follows:

"Fig. 7b which showed strong correlations between the shape parameters of these two approaches (r2=0.96) but a slight positive bias with positive groundwater contributions for the inferred evapotranspiration approach; 62% of watersheds overall and 86% of watersheds with a positive groundwater contribution have a higher shape parameter using the inferred evapotranspiration approach."

L578: do you mean Fig. 6? This should have been referring to figure 7b. We have corrected this in the revised manuscript.

L603: 100? Do you not mean 1000? Yes, this should be 1,000 not 100. This has been corrected.

L607: References needed here.

We have added two references here: [Budyko, 1974] and [Donohue et al., 2007].

Figure 9: General observation here: As the authors state, the analysis results that focus on the larger watersheds provides a better match to the idealized curves. Could this result be at least in part due to the major issues identified earlier in this review that highlight the weaknesses in the authors approach towards simulating the groundwater component of the water balance for small watersheds? We refer the referee back to our earlier response with respect to groundwater simulation in small watersheds, which we do not think is a limiting factor here. In the detailed validation provided in Maxwell and Condon [2016] strong performance was observed across large and small watershed and there was no systematic bias with respect to drainage area. Also, we would like to clarify that what we are showing here is that the results converge more closely around a single curve for larger watersheds and that there is more variability for smaller watersheds. This finding is consistent with the original work by Budyko [1974] that showed a better convergence around a single curve for basins larger than 10,000 km2 as well as subsequent research by Donohue et al. [2007] which explained the need to include more catchment specific effects as you move to smaller drainage areas. The reviewer is correct that uncertainty in local parameters can shift the points for individual watersheds; however, because we are looking at more than 30,000 watersheds here the overall result shown in Figure 9 is a function of the variability sampled at every scale. Furthermore, we are showing the effective precipitation approach here as an example, but as noted in the text, similar convergence behavior was observed for the other two approaches as well. Therefore, the convergence at larger scales is not a function of the approach used. This is also demonstrated in Figure 10 where you can see that the

**three approaches have different central tendencies but all follow a pattern of decreased variance with increased drainage area.**

L612: No doubt watershed characteristics are important, but what about limitations with the way groundwater flow in small watersheds is represented in the model, and the adequacy of the employed datasets for meeting such finely resolved objectives?

Again we refer the referee back to our earlier reply. We do not feel we have a specific limitation with respect to groundwater flow in small watersheds. While we agree that datasets are uncertain and could change individual point values, this would not change the overall point that variability between watersheds will be greater for smaller drainage areas than for larger areas. As noted in the manuscript, we have validated the model against more than one million observations points for the 1985 simulation period. The detailed analysis of these comparisons is presented in the supplemental information of *Maxwell and Condon* [2016]. While we refer the reader to the original publication for details on model validation, we would like to point out here that our results do not show a systematic bias with drainage area. For example, the comparison of simulated versus observed stream flows from *Maxwell and Condon* [2016] reproduced below does not show a systematic bias for smaller stream gauges.

---

## Author Comment (AC3) · 18 Nov 2016

**We have provided point by point responses below with author replies in bold.**

Overall, I think this paper has the potential to turn into a good contribution that elaborates the influence of groundwater on the Budyko Hypothesis. The paper does not seem to have a well-described objective. I did not see a set of research questions or hypotheses to be tested. All the results presented in the paper are based on a single water year simulation in the ParFlow model, which is a fairly short time scale to convincingly report and use any groundwater related modeled variables. As I tried to figure out what the objectives of this paper might be I kept asking myself the following questions. Is the idea to:

1) develop a conceptual model for incorporating the role of groundwater (GW) to the Budyko hypotheisis (BH)?
2) parameterize the contribution of GW in the BH by relating the w parameter in the Zhang equation to a GW variable that may be obtained from observations or models, which can be used as a simple model?
3) evaluate model results to see when a Budyko type behavior is generated in systems where GW cannot be neglected (e.g. Fig 4), by modifying the source of water in the axis of the BH plotting position?

**We thank the referee for their careful review of our work and their thoughtful suggestions. We agree that the intent of this work did not come through clearly enough in the original manuscript, and as a result some of our findings and our methodology choices were also not transparent. We have added the following text to the end of the introduction to clearly state the motivation for our work, our hypothesis and the goals for the paper. Additionally, we have made significant changes throughout the manuscript, especially the abstract and conclusions, to tie all of the discussion back to these goals.**

> *"…Groundwater observations sufficient to precisely characterize watershed storage changes are difficult to obtain and are not widely available. Therefore, it seems unlikely that groundwater storage calculations will be added into most Budyko analyses; more work is needed to understand the sensitivity of Budyko relationships to changes in storage and the implications of assuming of no storage changes without the ability to regularly verify this assumption.*
>
> *We have identified three main approaches to estimate evapotranspiration (E) in Budyko analysis: First, if it's not possible to measure E directly, it is commonly estimated as the difference between precipitation and river outflow in a basin. Second, some studies measure E directly using a variety of field methods. Third, as is the case with the more recent studies that seek to account for storage changes, observed E values are augmented with measurements of groundwater surface water exchanges to estimate the 'effective precipitation' that is available for surface processes (i.e. outflow and evapotranspiration).*
>
> *Here we hypothesize that storage changes will bias Budyko results in predictable ways, as has been indicated by previous studies, but that the direction of the bias will vary based on the way that evapotranspiration is handled within a study. We evaluate this hypothesis by comparing Budyko relationships generated following the three different approaches using the outputs of a physically based hydrologic model that directly simulates the integrated groundwater surface water system over a large spatial domain at high resolution. The three primary goals of our comparative analysis are as follows:*
>> *1. Evaluate the sensitivity of Budyko relationships to groundwater storage changes*
>> *2. Characterize systematic differences in the impact of storage changes on Budyko relationships*

3. *Illustrate variability between approaches across physical settings and spatial scales"*

It was never clear to the reader why three water balance conceptualizations were used and why w were calculated for all three of them (Figs 5, 6, 7). The authors need to state what their goals were

**The purpose of the three different approaches is to mimic the methods that are commonly used in the literature and to see how results would change as a function of the approach chosen. We hope that this focus comes through more clearly now with the text added to the introduction above. Also, we moved the description of these three approaches into a new sub-section in the methods section and further clarified our approach with the following summary added to the start of the methods section:**

> *"We use an integrated hydrologic model to simulate water and energy fluxes in both the surface and the subsurface. Here we apply a high resolution (1 km$^2$) simulation of the majority of the continental U.S. which covers more than 6 M km$^2$ and simulates hydrologic systems across a broad range of physical settings and storage change magnitudes. The model is driven using historical observed atmospheric forcings such as precipitation and temperature and provides gridded outputs of all water and energy fluxes throughout the system. We use simulated surface water flow, evapotranspiration and groundwater surface water exchanges to calculate Budyko relationships using three different approaches to estimate fluxes:*
>
> *1. Calculating evapotranspiration from simulated runoff and precipitation*
> *2. Using simulated evapotranspiration values directly*
> *3. Using simulated evapotranspiration values directly and taking into account storage changes.*
>
> *Differences between the approaches are compared with storage changes in each basin to evaluate the systematic impacts of these changes on Budyko relationships.*
>
> *The numerical modeling approach used here provides several important advantages for this type of analysis. Within the numerical framework, groundwater surface water exchanges for every watershed in the system are fully characterized. This guarantees perfect closure of the water balance and means that we can mimic all three approaches within a consistent numerical framework where storage changes are directly accounted for. Furthermore, because the goal is to understand differences between approaches and not to predict local Budyko parameters the key advantage here is the ability to evaluate physically realistic behavior across a variety of physical settings and spatial scales where groundwater can be fully accounted for. Within this context, it should also be noted that the focus is on how groundwater storage changes perturb relationships. Therefore, uncertainty in local model parameters is much less important than realistic simulation of physical interactions for a range of storage changes and aridity values within a controlled numerical framework."*

If one needs to improve the use of the BH for regions where GW can not be neglected, one could work with the original model inputs of observed P and Q, and calculated Ep, and parameterize w = f(GW, E/P, Ep/P) and use this w in the original model and test it .. In your case E would come from PArFLow. Apparently this does not seemed to be the objective of this paper, but I felt that Fig 6b came close to this idea but stopped there.. Finding w value for the indirect method (Fig 6a) did not make sense to me as E=P-Q won't give the "correct" ET and therefore why would you calculate w using this ET/P. Please better state what you objectives are.

**Again we hope that the modifications to the text have made our goals clearer. Our purpose is not to predict shape parameters but to provide a demonstration of the sensitivity of these parameters to both groundwater storage changes and the methodology chosen for a study. We agree that inferring evapotranspiration as P-Q won't give the 'correct' ET value. However, as this is a very common method used in Budyko analyses we feel it is important to understand how groundwater storage changes influence Budyko behavior when this approach is used. The scatter in Fig. 6 shows that it's not possible to predict shape parameters as a function of groundwater contributions alone. This is to be expected given the other watershed characteristics that have been shown by previous research to impact Budyko curves. However, the point of Fig. 7 is to show that for a given watershed the shift in shape parameter resulting from a non-zero groundwater contribution will follow regular patterns. It may not be feasible for most studies to incorporate groundwater surface water interactions into their Budyko analyses (especially for those studies that do not even have evapotranspiration measurements to use); we seek to demonstrate how shape parameters will shift results across a broad range of settings and groundwater contribution levels so that other studies can take this account when interpreting their results even if it's not possible to directly verify the no storage assumption with observations. We have added the following text to the conclusions emphasize this point:**

> *"These results also have implications for the myriad of studies that seek to relate shape parameters for Budyko curves to other watershed characteristics. The conceptual models shown here illustrate that groundwater contributions will shift points in consistent and predictable ways when other variables are held constant (i.e. if you apply a consistent groundwater contribution across the entire range of aridity values or consider the shift of a single point with a given aridity value). However, we use the results from our integrated hydrologic model to demonstrate that, within complex domains, groundwater surface water exchanges are spatially heterogeneous and depend on watershed characteristics such as aridity values that can also influence Budyko relationships. The scatter in Figs. 6 and 7 demonstrates that groundwater contributions cannot easily serve as an independent predictor of the shape of Budyko relationships. This also shows that in large comparative studies, the bias caused by groundwater surface water interactions may not be readily apparent because it will vary from watershed to watershed."*

Interpretation of Figure 4,5,6,7 need help. The paper does not sufficiently discuss the processes that lead to patterns in these figures.
**We appreciate the comment and upon review we agree that the discussion of these figures could be improved. We have significantly expanded the discussion around these figures (as well as the rest of the results) to more explicitly step through the relationships shown and what they mean.**

Abstracts lines 25, 26 what do you mean by best results? Best of what? Is the "best" ˇ represent better predicted water balance by the BH, modified in this study, against modeled water balance? Or did you develop a simpler model of water balance that gives consistent predictions with ParFlow?
**We agree that this language was vague. The abstract has been significantly revised in response to other comments and this text is no longer included.**

Modeling methods: In this study modeled data comes from ParFlow, which was used for only a single water year (1985), starting from a steady-state groundwater configuration. Obviously the question is – why would you use a single water year. I wonder if the system won't respond to this steady-state assumption when you start running the model with the actual climate forcing of 1985. The paper

mentions that PArFlow simulations were done for historical climate in CONUS. I wonder why the authors did not use the full length of simulations and evaluate the –mean annual water balance Twith ˘ GW contribution in the BH hypothesis, instead of just using a single year which I presume creates some rapid transient conditions in the beginning of the model run as the water table would respond to the 1985 forcing. Running the model with a historical climate forcing data and evaluating the long-term water balance with long-term-average estimated flux variables, including groundwater seems to be the logical way to go. I'm having a hard time accepting the justification of the use of a single water year. BH is ideal for long-term-aver water balance conditions as well. So logic tells me to use longer simulations.

**It's clear from this comment that some of the description of our modeling approach and decisions were unclear. When we said that we were using 'historical climate', we meant that we were using the observed historical climate data for water year 1985 not that we have completed a long term transient simulation over a longer historical climate period. We have added text to clarify that the historical climate forcings we used are only for the one-year simulation that was completed. Also, the referee is correct that there is some initial instability when we transition from the steady state-groundwater configuration to the transient ParFlow CLM simulation. We did take this into account and discarded some initialization before starting the one-year simulation presented here. We have also added text to clarify this point.**

**We also agree that the one-year simulation period is not ideal if our goal is to achieve a long-term water balance. However, as we hope has been clarified in this response and in the revised manuscript, our focus is on understanding the impact of groundwater surface water exchanges when long term equilibrium is either not the case or cannot be verified. We have added the following text to the methods section of the revised manuscript to be more explicit about this choice:**

> ***"The 1-year simulation presented here intentionally violates the steady state assumption. The purpose of our analysis is to evaluate the impact of net storage changes on Budyko relationships, therefore a steady-state simulation is not the goal. It can also be argued that storage changes will vary from year to year or depending on the multi-year period analyzed. The 1985 simulation year is not presented as a prediction of long-term storage variability, it is simply used to sample a range of groundwater surface water exchange across variable climates and physical settings. We present a general framework for understanding the impacts of storage changes in various Budyko formulations using water year 1985 as a representative example."***

In the methods the G term need to be more clearly explained in my opinion. In reading the paper I went back and forth a few times to make sense of what authors might have meant by G but I'm still not clear.. My intuition tells me that groundwater contribution would be the net volume of groundwater staying in the basin at the end of a water year.. I imagine G is not always a contribution as in some cases G may flow out of a basin in which case G will be a sink term. Your groundwater surface water exchange can practically be infiltration or saturation excess overland flow.. I'm not following what this definition means in the context of eq.(6). Your sign convention in the G/P plots should be explained.

**The groundwater contribution term is not a measure of groundwater storage or lateral groundwater fluxes; it represents the net exchange between the surface and the subsurface. In response to this comment we have added the following discussion to better clarify what is and isn't included in the groundwater contribution term (new text is underlined). Also, throughout the results section we have added text to remind the reader that a positive groundwater contribution means a net flux of water from the surface to the surface.**

*"There are multiple ways to estimate groundwater contributions within the model. Using gridded model outputs, the exchanges across the boundaries of every river cell could be summed to determine net contribution of groundwater to overland flow. Similarly, we could aggregate hourly changes in groundwater storage for every sub basin to determine total storage exchanges. Because we are interested in the net contribution of groundwater to streamflow and evapotranspiration for this analysis, we can take a simpler approach. Within our numeral framework we have guaranteed closure of the water balance for every watershed and therefore the net change in groundwater storage that contributes to the surface water budget is simply P - $Q_{out}$ – E, based on Eq. (6). When calculated this way, G encompasses the total groundwater surface water exchanges (i.e. changes in storage) required to support the simulated outflow and evapotranspiration. It should be noted that in this formulation G encompasses both exchanges between groundwater and surface water, which can be either positive fluxes from the surface to the subsurface or negative fluxes from subsurface to the surface, as well as changes in surface water storage. The assumption is that, over the annual simulation, changes in ponded water are small relative to groundwater surface water exchanges and so we refer to G as simply groundwater storage changes or groundwater contributions. We follow the convention that a positive groundwater contribution denotes water that is infiltrating from the land surface to the subsurface whereas a negative value indicates groundwater discharge which can either occur from groundwater supported E or baseflow contributions to streams.*

*This approach is focused solely on the net contribution of groundwater to the surface water budget. Nested systems of local and regional lateral groundwater flow are simulated within the model and previous work has evaluated spatial patterns and physical drivers of lateral groundwater imports and exports across the domain[Condon and Maxwell, 2015; Condon et al., 2015], as well as groundwater residence times [Maxwell et al., 2016]. Here we focus only on net exchanges with the surface that are relevant to the Budyko formulation. We do not need to quantify lateral exchanges in the subsurface directly for these purposes; however, it should be noted that the lateral redistribution of groundwater that occurs within the model is still vital to generating realistic groundwater configurations and supporting groundwater surface water exchanges."*

Line 319– I'm not following this para.. shouldn't a positive G mean that the watershed receives flux across its groundwater boundaries and a negative G indicates net export of GW to surrounding basins.. Water that infiltrates to subsurface would just increase the storage of GW wouldn't it.. This water may stay in the watershed or exported out.. seems like concepts are a bit miss-use here or not explained clearly.. Perhaps you use a Delta Storage term in 6 and 7 and explain these referring to the storage change etc.. hard to follow here..

**We refer the referee to the response to the previous comment. A positive G value indicates a net flux of water from the surface to the subsurface (i.e. a positive contribution to groundwater). G is not a measure of groundwater storage or lateral flow, it is simply reflecting the extent to which the surface water budget is perturbed by exchanges with the subsurface. We hope that the referee will find the modified discussion copied above to be easier to be more transparent.**

Line 353- I'm not clear how G was calculated.. Above you said you used eq (6).. here G/P>0 is interpreted as storage gain.. Headwater of Missouri should be recharging the system and therefore they are not GW exporters.. but the region below in ND and Nebraska area should be net exporters right..? so I was expecting to so G/P0 toward the middle of Missouri where it connects to Mississippi.. Please

better explain conceptual model. Please clarify– if G>0, there should be net input to the watershed and Effective precip should be P+G, and if G

**Again we refer the referee to the previous response and to the studies cited there that evaluate lateral flow patterns directly. As stated above, we have done work to evaluate groundwater imports and exports with this model, but the groundwater fraction presented here is not a direct measure of lateral groundwater flow between basins (although clearly the net exchanges we are using here will be supported by lateral groundwater flow in the model).**

Line 419.. I would not cite Istanbulluoglu et al., 2012 here. Istanbulluoglu et al showed the limitation of assuming ET=P-Q in the Budyko curve, and proposed to use ET= P-QDeltaS, where DeltaS is change in groundwater storage assuming no net export/import of GW.

**We agree; although Istanbulluoglu et al 2012 did use the inferred approach, their intent was to evaluate the limitations of this approach when storage changes occur. We agree that they should not be grouped with other studies that were not considering groundwater changes and the citation has been removed.**

Line 453..don't cite Istanbulluoglu et al., 2012 here

**This has also been corrected in the revised manuscript.**

Line 525.. Istanbulluoglu et al., 2012 used the inferred ET approach to show its limitations– not as the proposed method to calculate ET from P and Q. Incorporating the contribution of groundwater in the water balance equation to calculate ET led to a more consistent trend in the evapotranspiration ratio and aridity index.. See Figs 6a,b and Fig 7a,b,e,f. I think this paragraph should better summarize their results.

**We have expanded the summary of Istanbulluoglu et al. as follows (new text is underlined):**

> *"This point is also made by Istanbulluoglu et al. [2012] who evaluated the impact of groundwater storage changes on Budyko relationships using the inferred evapotranspiration approach and adjusting for storage changes using estimates from groundwater observations. They provide a similar conceptual model to Fig. 6d describing consistent shift within the Budyko space as a function of groundwater contribution. However, for the four basins in Nebraska that they evaluated they found a negative relationship between inferred evapotranspiration ratios and aridity. This was attributed to a strong negative correlation between groundwater contribution fraction and aridity index. In other words, for this subset of basins, they show that the resulting trend is controlled by the dependence of groundwater contribution on other watershed characteristics."*

**Works Cited:**

Condon, L. E., and R. M. Maxwell (2015), Evaluating the relationship between topography and groundwater using outputs from a continental-scale integrated hydrology model, Water Resources Research, n/a-n/a, doi: 10.1002/2014WR016774.

Condon, L. E., A. S. Hering, and R. M. Maxwell (2015), Quantitative assessment of groundwater controls across major US river basins using a multi-model regression algorithm, Advances in Water Resources, 82, 106-123, doi: http://dx.doi.org/10.1016/j.advwatres.2015.04.008.

Istanbulluoglu, E., T. Wang, O. M. Wright, and J. D. Lenters (2012), Interpretation of hydrologic trends from a water balance perspective: The role of groundwater storage in the Budyko hypothesis, Water Resources Research, 48(3), n/a-n/a, doi: 10.1029/2010WR010100.

Maxwell, R. M., L. E. Condon, S. J. Kollet, K. Maher, R. Haggerty, and M. M. Forrester (2016), The imprint of climate and geology on the residence times of groundwater, Geophysical Research Letters, 43(2), 701-708, doi: 10.1002/2015GL066916.

---

## Author Comment (AC4) · 18 Nov 2016

**We have provided point by point responses below with author replies in bold.**

The authors have used a physically based hydrological model to improve waterbudgeting at catchment scale. In particular, they have considered the original Budyko model as a reference and shown that by accounting for ground water inflow/outflow, water budgeting can be done more accurately. I really appreciate the authors' effort to undertake such an extensive numerical analysis. The article looks suitable for publication although I think a couple of key concerns the authors need to address.
**We thank the referee for their support of our work and appreciate their suggestions. We have provided detailed responses to each point below.**

1. Purpose of the study
 The authors need to elaborate on the usefulness of their study. The physically based hydrological model they are using has many parameters; they cannot take that model to a random ungauged catchment and predict its hydrological variables. On the other hand, the Budyko model is a universal deterministic model which can be applied to any ungauged catchment. It is thus not surprising that the multi-parameter model will perform better after calibration. I don't think their study is very informative unless they integrate a deterministic physically based hydrological model with the Budyko model to improve prediction.
**We agree with the limitations noted by the reviewer and would like to clarify that the purpose of our study is not to predict shape parameters or evapotranspiration in ungauged basins. Rather we are using the model as a controlled numerical experiment to demonstrated they ways that storage changes will influence Budyko relationships across a broad range of physical settings and for various experimental approaches. The purpose here is to better inform other studies that seek to attribute variance in the Budyko space to physical watershed characteristics but that may be lacking the required data to account for groundwater surface water interactions.  We have revised the manuscript significantly to better focus on these goals throughout and especially in the abstract introduction and conclusions.  For example, we have added the following text to the end of the introduction to clarify this point:**

> *"…Groundwater observations sufficient to precisely characterize watershed storage changes are difficult to obtain and are not widely available. Therefore, it seems unlikely that groundwater storage calculations will be added into most Budyko analyses; more work is needed to understand the sensitivity of Budyko relationships to changes in storage and the implications of assuming of no storage changes without the ability to regularly verify this assumption.*

> *We have identified three main approaches to estimate evapotranspiration (E) in Budyko analysis: First, if it's not possible to measure E directly, it is commonly estimated as the difference between precipitation and river outflow in a basin.  Second, some studies measure E directly using a variety of field methods.  Third, as is the case with the more recent studies that seek to account for storage changes, observed E values are augmented with measurements of groundwater surface water exchanges to estimate the 'effective precipitation' that is available for surface processes (i.e. outflow and E).*

> *Here we hypothesize that storage changes will bias Budyko results in predictable ways, as has been indicated by previous studies, but that the direction of the bias will vary based on the way that evapotranspiration is handled within a study.  We evaluate this hypothesis by comparing Budyko relationships generated following the three different approaches using the*

*outputs of a physically based hydrologic model that directly simulates the integrated groundwater surface water system over a large spatial domain at high resolution. The three primary goals of our comparative analysis are as follows:*
1. *Evaluate the sensitivity of Budyko relationships to groundwater storage changes*
2. *Characterize systematic differences in the impact of storage changes on Budyko relationships*
3. *Illustrate variability between approaches across physical settings and spatial scales"*

2. Clarity of presentation.
It is quite hard to follow what the authors are saying at many places. In my opinion, the presentation needs to be simple. If the authors' objective is to show how the physically based hydrological model is doing a better job at water budgeting, they need to focus on that part more. There is not a single figure showing a direct comparison between prediction by the physically based hydrological model and that by the Budyko model... Terms need to be defined prior to their usage. For example, in Line 27 the authors are talking about Budyko curve parameters. The authors are actually talking about Fu model's parameters (Budyko model does not have any parameter).
**We have significantly revised the manuscript following this comment, as well as the comments from other referees, to try to improve clarity. We have revised the abstract and this text is no longer included. As noted above we are now much more explicit about our goals in the introduction. Also, we reorganized the methods section to include all of the details of the three approaches, expanded the discussion of the results figures to be more descriptive and we shortened the conclusions section to remove redundant material and focus back on the original goals of the paper.**

---

## Author Response (AR2)

**SYRACUSE UNIVERSITY**

[Figure]

*Laura E. Condon*
*Assistant Professor*
*Department of Civil and Environmental Engineering*
*Syracuse University*
*151 Link Hall*
*Syracuse NY, 12344*
*lecondon@syr.edu*

February 2, 2017

Dear Dr. Bierkens,

We respectfully resubmit our manuscript titled "Systematic shifts in Budyko relationships caused by groundwater storage changes" by Laura Condon and Reed Maxwell. Per the suggestion of reviewer 1 we have now clarified that the small number of points that fall to the left of the energy limit line on figure 5 are caused by the treatment of atmospheric stability in calculation of potential evapotranspiration. We think that this clarification will address their concern and we look forward to hearing your decision. Should you need any additional information please do not hesitate to contact me.

Sincerely,

Laura Condon

Response to Referee 1:
**We thank the referee for reviewing our manuscript a second time and for their positive comments. We have responded to their comment below in bold.**

This revised MS makes important issues clearer than previous; the only concerned thing is that some points (watersheds) shift towards left away from the energy limit line on both Fig. 5a and Fig. 5c, and the authors probably need to give some explainations.

**We agree with the referee that these points could use some explanation. In a very small number of basins (less than 20 out of more than 3,000) we get simulated ET that is higher than our projected potential ET. This is because of the treatment of atmospheric stability in the calculation of potential ET. To clarify this point, we've added the following discussion to the description of the potential ET calculations"**

*"This approach is designed to be consistent with the CLM simulation of ET but is slightly simplified because it does not evaluate atmospheric stability (refer to Jefferson and Maxwell [2015] for a detailed comparison of different formulations)." (revised MS lines 437-440)*

**We have also included this in the discussion of Figure 5 as follows:**

*"There are also a small number of points (less than 20) in subplots a and c that fall to the left of the energy limit line; this behavior results from the treatment of atmospheric stability in the Ep calculation." (revised MS lines 590-592)*

[revised manuscript text omitted]